# Hierarchical Randomized Smoothing

**Yan Scholten**[1], **Jan Schuchardt**[1], **Aleksandar Bojchevski**[2] **& Stephan Günnemann**[1]
`{y.scholten, j.schuchardt, s.guennemann}@tum.de, a.bojchevski@uni-koeln.de`
[1]Dept. of Computer Science & Munich Data Science Institute, Technical University of Munich
[2]University of Cologne, Germany

## Abstract

Real-world data is complex and often consists of objects that can be decomposed into multiple entities (e.g. images into pixels, graphs into interconnected nodes). Randomized smoothing is a powerful framework for making models provably robust against small changes to their inputs – by guaranteeing robustness of the majority vote when randomly adding noise before classification. Yet, certifying robustness on such complex data via randomized smoothing is challenging when adversaries do not arbitrarily perturb entire objects (e.g. images) but only a subset of their entities (e.g. pixels). As a solution, we introduce hierarchical randomized smoothing: We partially smooth objects by adding random noise only on a randomly selected subset of their entities. By adding noise in a more targeted manner than existing methods we obtain stronger robustness guarantees while maintaining high accuracy. We initialize hierarchical smoothing using different noising distributions, yielding novel robustness certificates for discrete and continuous domains. We experimentally demonstrate the importance of hierarchical smoothing in image and node classification, where it yields superior robustness-accuracy trade-offs. Overall, hierarchical smoothing is an important contribution towards models that are both – certifiably robust to perturbations and accurate.[1]

## 1 Introduction

Machine learning models are vulnerable to small input changes. Consequently, their performance is typically better during development than in the real world where noise, incomplete data and adversaries are present. To address this problem, robustness certificates constitute useful tools that provide provable guarantees for the stability of predictions under worst-case perturbations, enabling us to analyze and guarantee robustness of our classifiers in practice.

Randomized smoothing (Lecuyer et al., 2019; Cohen et al., 2019) is a powerful and highly flexible framework for making arbitrary classifiers certifiably robust by averaging out small discontinuities in the underlying model. This is achieved by guaranteeing robustness of the majority vote when randomly adding noise to a model's input before classification. This typically yields a robustness-accuracy trade-off – more noise increases certifiable robustness but also reduces clean accuracy.

Recent adversarial attacks against image classifiers (Su et al., 2019; Croce and Hein, 2019; Modas et al., 2019) and graph neural networks (Ma et al., 2020) further motivate the need for more flexible robustness certificates against threat models where adversaries are bounded by both: the number of perturbed entities and perturbation strength. For example in social networks, adversaries typically cannot control entire graphs, but the perturbation of the controlled nodes may be (imperceptibly) strong.

When adversaries perturb only a subset of all entities, existing smoothing-based approaches sacrifice robustness over accuracy (or vice versa): They either add noise to entire objects (e.g. images/graphs) or ablate entire entities (e.g. pixels/nodes) even if only a small subset is perturbed and requires noising.

---

[1]Project page: `https://www.cs.cit.tum.de/daml/hierarchical-smoothing`

37th Conference on Neural Information Processing Systems (NeurIPS 2023).

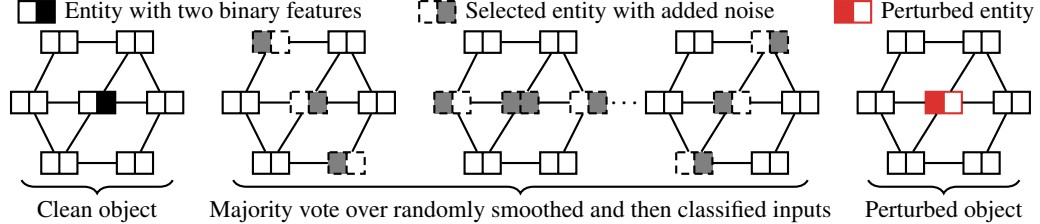

Figure 1: Hierarchical randomized smoothing: We first select a subset of all entities and then add noise to the selected entities only. We achieve stronger robustness guarantees while still maintaining high accuracy – especially when adversaries can only perturb a subset of all entities. For example in social networks, adversaries typically control only a subset of all nodes in the entire graph.

The absence of appropriate tools for decomposable data prevents us from effectively analyzing the robustness of models such as image classifiers or graph neural networks – raising the research question: *How can we provably guarantee robustness under adversaries that perturb only a subset of all entities?* We propose hierarchical randomized smoothing: By only adding noise on a randomly selected subset of all entities we achieve stronger robustness guarantees while maintaining high accuracy (Figure 1).

Our hierarchical smoothing framework is highly flexible and can integrate the whole suite of smoothing distributions for adding random noise to the input, provided that the corresponding certificates exist. Notably, our certificates apply to a wide range of models, domains and tasks and are at least as efficient as the certificates for the underlying smoothing distribution.

We instantiate hierarchical smoothing with well-established smoothing distributions (Cohen et al., 2019; Bojchevski et al., 2020; Levine and Feizi, 2020b; Scholten et al., 2022), yielding novel robustness certificates for discrete and continuous domains. Experimentally, hierarchical smoothing significantly expands the Pareto-front when optimizing for accuracy and robustness, revealing novel insights into the reliability of image classifiers and graph neural networks (GNNs).

In short, our main contributions are:

- We propose a novel framework for strong robustness certificates for complex data where objects (e.g. images or graphs) can be decomposed into multiple entities (e.g. pixels or nodes).
- We instantiate our framework with well-established smoothing distributions, yielding novel robustness certificates for discrete and continuous domains.
- We demonstrate the importance of hierarchical smoothing in image and node classification, where it significantly expands the Pareto-front with respect to robustness and accuracy.

## 2  Related work

Despite numerous research efforts, the vulnerability of machine learning models against adversarial examples remains an open research problem (Hendrycks et al., 2021). Various tools emerged to address adversarial robustness: Adversarial attacks and defenses allow to evaluate and improve robustness empirically (Szegedy et al., 2014; Goodfellow et al., 2015; Carlini et al., 2019; Croce and Hein, 2019). While attacks provide an upper bound on the adversarial robustness of a model, robustness certificates provide a provable lower bound. The development of strong certificates is challenging and constitutes a research field with many open questions (Li et al., 2020).

**Robustness certification via randomized smoothing.** Randomized smoothing (Lecuyer et al., 2019; Cohen et al., 2019) represents a framework for making arbitrary classifiers certifiably robust while scaling to deeper architectures. There are two main approaches discussed in the literature: Additive noise certificates add random noise to continuous (Cohen et al., 2019) or discrete (Lee et al., 2019) input data. Ablation certificates "mask" out parts of the input to hide the potentially perturbed information (Levine and Feizi, 2020b). Since the input is masked, ablation certificates can be applied to both continuous and discrete data. We are the first to combine and generalize additive noise and ablation certificates into a novel hierarchical framework: We first select entities to ablate, but instead of ablating we add noise to them. By only partially adding noise to the input our framework is more flexible and allows us to better control the robustness-accuracy trade-off under our threat model.

Further research directions and orthogonal to ours include derandomized (deterministic) smoothing (Levine and Feizi, 2020a, 2021; Horváth et al., 2022b; Scholten et al., 2022), input-dependent smoothing (Súkeník et al., 2022), and gray-box certificates (Mohapatra et al., 2020; Levine et al., 2020; Schuchardt and Günnemann, 2022; Scholten et al., 2022; Schuchardt et al., 2023a).

**Robustness-accuracy trade-off.** The robustness-accuracy trade-off has been extensively discussed in the literature (Tsipras et al., 2019; Raghunathan et al., 2019; Zhang et al., 2019; Yang et al., 2020b; Xie et al., 2020), including works for graph data (Gosch et al., 2023b). In randomized smoothing the parameters of the smoothing distribution allow to trade robustness against accuracy (Cohen et al., 2019; Mohapatra et al., 2020; Wu et al., 2021) where more noise means higher robustness but lower accuracy. Our framework combines two smoothing distributions while introducing parameters for both, which allows us to better control the robustness-accuracy trade-off under our threat model since we can control both – the number of entities to smooth and the magnitude of the noise. Further advances in the trade-off in randomized smoothing include the work of Levine and Feizi (2020a) against patch-attacks, Schuchardt et al. (2021, 2023b) for collective robustness certification, and Scholten et al. (2022) for graph neural networks. Other techniques include consistency regularization (Jeong and Shin, 2020), denoised smoothing (Salman et al., 2020) and ensemble techniques (Müller et al., 2021; Horváth et al., 2022a,c). Notably, our method is orthogonal to such enhancements since we derive certificates for a novel, flexible smoothing distribution to better control such trade-offs.

**Robustness certificates for GNNs.** Research on GNN certification is still at an early stage (Günnemann, 2022), most works focus on heuristic defenses rather than provable robustness guarantees. But heuristics cannot guarantee robustness and may be broken by stronger attacks in the future (Mujkanovic et al., 2022). Most certificates are limited to specific architectures (Zügner and Günnemann, 2020; Jin et al., 2020; Bojchevski and Günnemann, 2019; Zügner and Günnemann, 2019). Randomized smoothing is a more flexible framework for certifying arbitrary models while only accessing their forward pass. We compare our method to additive noise certificates for sparse data by Bojchevski et al. (2020), and to ablation certificates by Scholten et al. (2022) that ablate all node features to derive certificates against strong adversaries that control entire nodes. While we implement certificates against node-attribute perturbations, our framework can in principle integrate smoothing-based certificates against graph-structure perturbations as well.

# 3 Preliminaries and background

Our research is focused on data where objects can be decomposed into entities. W.l.o.g. we represent such objects as matrices where rows represent entities. We assume a classification task on a discrete or continuous $(N \times D)$-dimensional space $\mathcal{X}^{N \times D}$ (e.g. $\mathcal{X} = \{0, 1\}$ or $\mathcal{X} = \mathbb{R}$). We propose certificates for classifiers $f : \mathcal{X}^{N \times D} \to \mathcal{Y}$ that assign matrix $\boldsymbol{X} \in \mathcal{X}^{N \times D}$ a label $y \in \mathcal{Y} = \{1, \dots, C\}$.[2] We model *evasion* settings, i.e. adversaries perturb the input at inference. We seek to verify the robustness $f(\boldsymbol{X}) = f(\tilde{\boldsymbol{X}})$ for perturbed $\tilde{\boldsymbol{X}}$ from a threat model of admissible perturbations:

$$\mathcal{B}_{p,\epsilon}^r(\mathbf{X}) \triangleq \left\{ \tilde{\mathbf{X}} \in \mathcal{X}^{N \times D} \mid \sum_{i=1}^{N} \mathbb{1}[\mathbf{x}_i \neq \tilde{\mathbf{x}}_i] \leq r, \, ||\text{vec}(\boldsymbol{X} - \tilde{\boldsymbol{X}})||_p \leq \epsilon \right\}$$

where $\mathbb{1}$ denotes an indicator function, $|| \cdot ||_p$ a $\ell_p$-vector norm and $\text{vec}(\cdot)$ the matrix vectorization. For example, $||\text{vec}(\cdot)||_2$ is the Frobenius-norm. Intuitively, we model adversaries that control up to $r$ rows in the matrix, and whose perturbations are bounded by $\epsilon$ under a fixed $\ell_p$-norm.

## 3.1 Randomized smoothing framework

Our certificates build on the randomized smoothing framework (Lecuyer et al., 2019; Li et al., 2019; Cohen et al., 2019). Instead of certifying a *base classifier* $f$, one constructs a *smoothed classifier* $g$ that corresponds to the majority vote prediction of $f$ under a randomized smoothing of the input. We define the smoothed classifier $g : \mathcal{X}^{N \times D} \to \mathcal{Y}$ for a smoothing distribution $\mu_{\boldsymbol{X}}$ as:

$$g(\boldsymbol{X}) \triangleq \underset{y \in \mathcal{Y}}{\arg\max} \, p_{\boldsymbol{X},y} \qquad p_{\boldsymbol{X},y} \triangleq \Pr_{\boldsymbol{W} \sim \mu_{\boldsymbol{X}}} [f(\boldsymbol{W}) = y]$$

where $p_{\boldsymbol{X},y}$ is the probability of predicting class $y$ for input $\boldsymbol{X}$ under the smoothing distribution $\mu_{\boldsymbol{X}}$. Let $y^* \triangleq g(\boldsymbol{X})$ further denote the majority vote of $g$, i.e. the most likely prediction for the clean $\boldsymbol{X}$.

---

[2]Multi-output classifiers can be treated as multiple independent single-output classifiers.

To derive robustness certificates for the smoothed classifier $g$ we have to show that $y^*=g(\boldsymbol{X})=g(\tilde{\boldsymbol{X}})$ for clean matrix $\boldsymbol{X}$ and any perturbed matrix $\tilde{\boldsymbol{X}} \in \mathcal{B}^r_{p,\epsilon}(\boldsymbol{X})$. This is the case if the probability to classify any perturbed matrix $\tilde{\boldsymbol{X}}$ as $y^*$ is still larger than for all other classes, that is if $p_{\tilde{\boldsymbol{X}},y^*} > 0.5$. However, randomized smoothing is designed to be flexible and does not assume anything about the underlying base classifier, i.e. all we know about $f$ is the probability $p_{\boldsymbol{X},y}$ for the clean matrix $\boldsymbol{X}$. We therefore derive a lower bound $\underline{p_{\tilde{\boldsymbol{X}},y}} \leq p_{\tilde{\boldsymbol{X}},y}$ under a worst-case assumption: We consider the least robust classifier among all classifiers with the same probability $p_{\boldsymbol{X},y}$ of classifying $\boldsymbol{X}$ as $y$:

$$\underline{p_{\tilde{\boldsymbol{X}},y}} \quad \triangleq \quad \min_{h \in \mathbb{H}} \Pr_{\boldsymbol{W} \sim \mu_{\tilde{\boldsymbol{X}}}} [h(\boldsymbol{W}) = y] \quad s.t. \quad \Pr_{\boldsymbol{W} \sim \mu_{\boldsymbol{X}}} [h(\boldsymbol{W}) = y] = p_{\boldsymbol{X},y} \tag{1}$$

where $\mathbb{H} \triangleq \{h : \mathcal{X}^{N \times D} \to \mathcal{Y}\}$ is the set of possible classifiers with $f \in \mathbb{H}$. To ensure that the smoothed classifier is robust to the entire treat model we have to guarantee $\min_{\tilde{\boldsymbol{X}} \in \mathcal{B}(\boldsymbol{X})} \underline{p_{\tilde{\boldsymbol{X}},y^*}} > 0.5$. One can also derive tighter multi-class certificates, for which we refer to Appendix D for conciseness.

**Probabilistic certificates.** Since computing $p_{\boldsymbol{X},y}$ exactly is challenging in practice, one estimates it using Monte-Carlo samples from $\mu_{\boldsymbol{X}}$ and bounds $p_{\boldsymbol{X},y}$ with confidence intervals (Cohen et al., 2019). The final certificates are probabilistic and hold with an (arbitrarily high) confidence level of $1 - \alpha$.

### 3.2 Neyman-Pearson Lemma: The foundation of randomized smoothing certificates

So far we introduced the idea of robustness certification using randomized smoothing, but to compute certificates one still has to solve Equation 1. To this end, Cohen et al. (2019) show that the minimum of Equation 1 can be derived using the Neyman-Pearson ("NP") Lemma (Neyman and Pearson, 1933). Intuitively, the least robust model classifies those $\boldsymbol{W}$ as $y$ for which the probability of sampling $\boldsymbol{W}$ around the clean $\boldsymbol{X}$ is high, but low around the perturbed $\tilde{\boldsymbol{X}}$:

**Lemma 1** (Neyman-Pearson lower bound). *Given $\boldsymbol{X}, \tilde{\boldsymbol{X}} \in \mathcal{X}^{N \times D}$, distributions $\mu_{\boldsymbol{X}}, \mu_{\tilde{\boldsymbol{X}}}$, class label $y \in \mathcal{Y}$, probability $p_{\boldsymbol{X},y}$ and the set $S_\kappa \triangleq \{\boldsymbol{W} \in \mathcal{X}^{N \times D} : \mu_{\tilde{\boldsymbol{X}}}(\boldsymbol{W}) \leq \kappa \cdot \mu_{\boldsymbol{X}}(\boldsymbol{W})\}$, we have*

$$\underline{p_{\tilde{\boldsymbol{X}},y}} = \Pr_{\boldsymbol{W} \sim \mu_{\tilde{\boldsymbol{X}}}} [\boldsymbol{W} \in S_\kappa] \quad \text{with } \kappa \in \mathbb{R}_+ \text{ s.t.} \quad \Pr_{\boldsymbol{W} \sim \mu_{\boldsymbol{X}}} [\boldsymbol{W} \in S_\kappa] = p_{\boldsymbol{X},y}$$

*Proof.* See (Neyman and Pearson, 1933) and (Cohen et al., 2019).

### 3.3 Existing smoothing distributions for discrete and continuous data

Our hierarchical smoothing framework builds upon certificates for smoothing distributions that apply noise independently per dimension (Lecuyer et al., 2019; Cohen et al., 2019; Lee et al., 2019). Despite being proposed for vector data $\boldsymbol{x} \in \mathcal{X}^D$, these certificates have been generalized to matrix data by applying noise independently on all matrix entries (Cohen et al., 2019; Bojchevski et al., 2020; Chu et al., 2022; Schuchardt and Günnemann, 2022). Thus, given matrix data we define the density $\mu_{\boldsymbol{X}}(\boldsymbol{W}) \triangleq \prod_{i=1}^N \prod_{j=1}^D \mu_{\boldsymbol{X}_{ij}}(\boldsymbol{W}_{ij})$ given the density of an existing smoothing distribution $\mu$.[3]

**Gaussian randomized smoothing.** For continuous data $\mathcal{X} = \mathbb{R}$, Cohen et al. (2019) derive the first tight certificate for the $\ell_2$-threat model. Given an isotropic Gaussian smoothing distribution $\mu_{\boldsymbol{x}}(\boldsymbol{w}) = \mathcal{N}(\boldsymbol{w}|\boldsymbol{x}, \sigma^2 \boldsymbol{I})$, Cohen et al. (2019) show that the optimal value of Equation 1 is given by $\underline{p_{\tilde{\boldsymbol{x}},y}} = \Phi\left(\Phi^{-1}(p_{\boldsymbol{x},y}) - \frac{||\boldsymbol{\delta}||_2}{\sigma}\right)$, where $\Phi$ denotes the CDF of the normal distribution and $\boldsymbol{\delta} \triangleq \boldsymbol{x} - \tilde{\boldsymbol{x}}$.

**Randomized smoothing for discrete data.** Lee et al. (2019) derive tight certificates for the $\ell_0$-threat model. Certificates for discrete domains are in general combinatorial problems and computationally challenging. To overcome this, Lee et al. (2019) use the NP-Lemma for discrete random variables (Tocher, 1950) and show that the minimum in Equation 1 can be computed with a linear program (LP):

**Lemma 2** (Discrete Neyman-Pearson lower bound). *Partition the input space $\mathcal{X}^D$ into disjoint regions $\mathcal{R}_1, \ldots, \mathcal{R}_I$ of constant likelihood ratio $\mu_{\boldsymbol{x}}(\boldsymbol{w})/\mu_{\tilde{\boldsymbol{x}}}(\boldsymbol{w}) = c_i \in \mathbb{R} \cup \{\infty\}$ for all $\boldsymbol{w} \in \mathcal{R}_i$. Define $\tilde{\boldsymbol{r}}_i \triangleq \Pr_{\boldsymbol{w} \sim \mu_{\tilde{\boldsymbol{x}}}}(\boldsymbol{w} \in \mathcal{R}_i)$ and $\boldsymbol{r}_i \triangleq \Pr_{\boldsymbol{w} \sim \mu_{\boldsymbol{x}}}(\boldsymbol{w} \in \mathcal{R}_i)$. Then Equation 1 is equivalent to the linear program $\underline{p_{\tilde{\boldsymbol{x}},y}} = \min_{\boldsymbol{h} \in [0,1]^I} \boldsymbol{h}^T \tilde{\boldsymbol{r}}$ s.t. $\boldsymbol{h}^T \boldsymbol{r} = p_{\boldsymbol{x},y}$, where $\boldsymbol{h}$ represents the worst-case classifier.*

Proof in (Tocher, 1950). The LP can be solved efficiently using a greedy algorithm that consumes regions with larger likelihood ratio first (Lee et al., 2019). If the number of regions is small and $\boldsymbol{r}_i, \tilde{\boldsymbol{r}}_i$ can be computed efficiently, robustness certificates for discrete data become computationally feasible.

---

[3] We write $\mu(\boldsymbol{Z})$ when referring to the density of the distribution $\mu$.

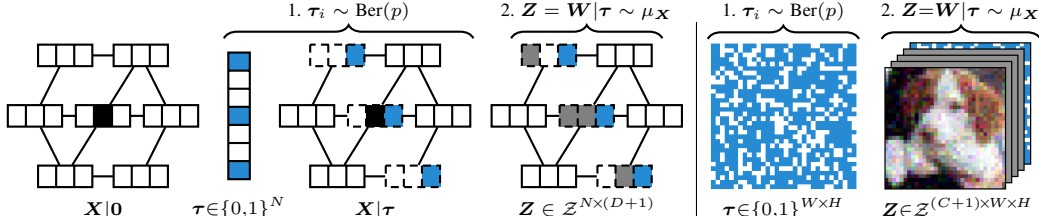

(a) For graphs we extend the feature matrix by an additional column that indicates which nodes in the graph to add noise to.

(b) For images we add an additional channel indicating the pixels to smooth.

Figure 2: We derive flexible and efficient robustness certificates for hierarchical randomized smoothing by certifying classifiers on a higher-dimensional space where the indicator $\boldsymbol{\tau}$ is added to the data.

## 4 Hierarchical smoothing distribution

We aim to guarantee robustness against adversaries that perturb a subset of rows of a given matrix $\boldsymbol{X}$. To achieve this we introduce hierarchical randomized smoothing: We partially smooth matrices by adding random noise on a randomly selected subset of their rows only. By adding noise in a more targeted manner than existing methods we will obtain stronger robustness while maintaining high accuracy. We describe sampling from the hierarchical (mixture) distribution $\Psi_{\boldsymbol{X}}$ as follows:

First, we sample an indicator vector $\boldsymbol{\tau} \in \{0,1\}^N$ from an **upper-level smoothing distribution** $\phi(\boldsymbol{\tau})$ that indicates which rows to smooth. Specifically, we draw each element of $\boldsymbol{\tau}$ independently from the same Bernoulli distribution $\boldsymbol{\tau}_i \sim \text{Ber}(p)$, where $p$ denotes the probability for smoothing rows. Second, we sample a matrix $\boldsymbol{W}$ by using a **lower-level smoothing distribution** $\mu$ (depending on domain and threat model) to add noise on the elements of the selected rows only. We define the overall density of this hierarchical smoothing distribution as $\Psi_{\boldsymbol{X}}(\mathbf{W}, \boldsymbol{\tau}) \triangleq \mu_{\boldsymbol{X}}(\boldsymbol{W}|\boldsymbol{\tau})\phi(\boldsymbol{\tau})$ with:

$$\mu_{\boldsymbol{X}}(\boldsymbol{W}|\boldsymbol{\tau}) \triangleq \prod_{i=1}^{N} \mu_{\mathbf{x}_i}(\mathbf{w}_i|\boldsymbol{\tau}_i) \qquad \text{for} \qquad \mu_{\boldsymbol{x}_i}(\mathbf{w}_i|\boldsymbol{\tau}_i) \triangleq \begin{cases} \mu_{\boldsymbol{x}_i}(\mathbf{w}_i) & \text{if } \boldsymbol{\tau}_i = 1 \\ \delta(\boldsymbol{w}_i - \boldsymbol{x}_i) & \text{if } \boldsymbol{\tau}_i = 0 \end{cases}$$

where $N$ is the number of rows and $\delta$ the Dirac delta. In the case of a discrete lower-level smoothing distribution, $\Psi_{\boldsymbol{X}}$ is a probability mass function and $\delta$ denotes the Kronecker delta $\delta_{\boldsymbol{w}_i, \boldsymbol{x}_i}$.

## 5 Provable robustness certificates for hierarchical randomized smoothing

We develop certificates by deriving a condition that guarantees $g(\boldsymbol{X}) = g(\tilde{\boldsymbol{X}})$ for clean $\boldsymbol{X}$ and any perturbed $\tilde{\boldsymbol{X}} \in \mathcal{B}_{p,\epsilon}^r(\mathbf{X})$. Certifying robustness under hierarchical smoothing is challenging, especially if we want to make use of certificates for the lower-level smoothing distribution $\mu$ without deriving the entire certificate from scratch. Moreover, Lemma 1 can be intractable in discrete domains where certification involves combinatorial problems. We are interested in certificates that are (**1**) **flexible** enough to make use of existing smoothing-based certificates, and (**2**) **efficient** to compute.

Our main idea to achieve both – flexible and efficient certification under hierarchical smoothing – is to embed the data into a higher-dimensional space $\boldsymbol{W} \hookrightarrow \mathcal{Z}^{N \times (D+1)}$ by appending the indicator $\boldsymbol{\tau}$ as an additional column: $\boldsymbol{Z} \triangleq \boldsymbol{W}|\boldsymbol{\tau}$ ("|" denotes concatenation, see Figure 2). We construct a new base classifier that operates on this higher-dimensional space, e.g. by ignoring $\boldsymbol{\tau}$ and applying the original base classifier to $\boldsymbol{W}$. In our experiments we also train the classifiers directly on the extended data.

Certifying the smoothed classifier on this higher-dimensional space simplifies the certification: By appending $\boldsymbol{\tau}$ to the data the supports of both distributions $\Psi_{\boldsymbol{X}}$ and $\Psi_{\tilde{\boldsymbol{X}}}$ differ, and they intersect only for those $\boldsymbol{Z}$ for which all perturbed rows are selected by $\boldsymbol{\tau}$ (see Figure 3). This allows the upper-level certificate to separate clean from perturbed rows and to delegate robustness guarantees for perturbed rows to an existing certificate for the lower-level $\mu$. We further elaborate on why this concatenation is necessary in Appendix H. In the following we show robustness certification under this hierarchical smoothing mainly involves (**1**) transforming the observed label probability $p_{\boldsymbol{X},y}$, (**2**) computing an existing certificate for the transformed $p_{\boldsymbol{X},y}$, and (**3**) transforming the resulting lower bound $\underline{p_{\tilde{\boldsymbol{X}},y}}$. We provide pseudo-code for the entire certification in Appendix C.

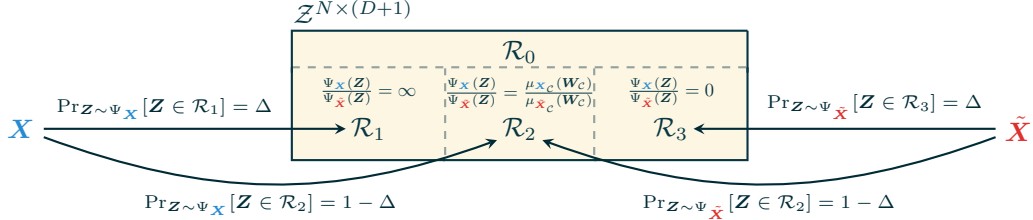

Figure 3: Overview of the disjoint regions and probabilities to sample $\boldsymbol{Z}$ of each region. The absence of arrows into specific regions indicates a probability of zero. Only $\boldsymbol{Z}$ in region $\mathcal{R}_2$ can be sampled from both distributions $\Psi_{\boldsymbol{X}}$ and $\Psi_{\tilde{\boldsymbol{X}}}$. The likelihood ratios visualize the proof of Theorem 1.

## 5.1 Point-wise robustness certificates for hierarchical randomized smoothing

Before providing certificates against the entire threat model we first derive point-wise certificates, i.e. we consider an arbitrary but fixed $\tilde{\boldsymbol{X}} \in \mathcal{B}_{p,\epsilon}^r(\boldsymbol{X})$ and show robustness $g(\boldsymbol{X}) = g(\tilde{\boldsymbol{X}})$ by deriving a lower bound $\underline{p_{\tilde{\boldsymbol{X}},y}} \le p_{\tilde{\boldsymbol{X}},y}$. The smoothed classifier $g$ is certifiably robust w.r.t. $\tilde{\boldsymbol{X}}$ if $\underline{p_{\tilde{\boldsymbol{X}},y}} > 0.5$. We derive a lower bound by using the discrete NP-Lemma on the upper-level $\phi$, which allows us to delegate robustness guarantees to the lower-level $\mu$ on the perturbed rows only.

Let $\mathcal{C} \triangleq \{i : \boldsymbol{x}_i \ne \tilde{\boldsymbol{x}}_i\}$ denote the rows in which $\boldsymbol{X}$ and $\tilde{\boldsymbol{X}}$ differ. To apply the discrete NP-Lemma we have to partition the space $\mathcal{Z}^{N \times (D+1)}$ into disjoint regions (Lemma 2). The upper-level distribution allows us define the following four regions (Figure 3): In region $\mathcal{R}_2$ all perturbed rows are selected ($\forall i \in \mathcal{C} : \boldsymbol{\tau}_i = 1$). In regions $\mathcal{R}_1$ and $\mathcal{R}_3$ at least one perturbed row is not selected when sampling from $\Psi_{\boldsymbol{X}}$ and $\Psi_{\tilde{\boldsymbol{X}}}$, respectively. Region $\mathcal{R}_0$ contains $\boldsymbol{Z}$ that cannot be sampled from either distribution.

**Example for matrices.** Consider the following example for matrix data to see the existence of four different regions. Define $\boldsymbol{X} \triangleq (\boldsymbol{x}_1|\boldsymbol{x}_2|\boldsymbol{x}_3)^T$ and $\tilde{\boldsymbol{X}} \triangleq (\boldsymbol{x}_1|\tilde{\boldsymbol{x}}_2|\boldsymbol{x}_3)^T$ (second row in $\tilde{\boldsymbol{X}}$ is perturbed). Now consider a smoothed row $\boldsymbol{w}$ with $\boldsymbol{w} \ne \boldsymbol{x}_2$ and $\boldsymbol{w} \ne \tilde{\boldsymbol{x}}_2$ that we can sample from both lower-level distributions $\mu_{\boldsymbol{x}_2}(\boldsymbol{w}) > 0$ and $\mu_{\tilde{\boldsymbol{x}}_2}(\boldsymbol{w}) > 0$. Then we can define four examples, one for each region:

$$
\begin{array}{cccc}
\boldsymbol{Z}_0 \in \mathcal{R}_0 & \boldsymbol{Z}_1 = \boldsymbol{X}|0 \in \mathcal{R}_1 & \boldsymbol{Z}_2 \in \mathcal{R}_2 & \boldsymbol{Z}_3 = \tilde{\boldsymbol{X}}|0 \in \mathcal{R}_3 \\
\begin{pmatrix} -\boldsymbol{x}_1- & 0 \\ -\boldsymbol{w}- & 0 \\ -\boldsymbol{x}_3- & 0 \end{pmatrix} & \begin{pmatrix} -\boldsymbol{x}_1- & 0 \\ -\boldsymbol{x}_2- & 0 \\ -\boldsymbol{x}_3- & 0 \end{pmatrix} & \begin{pmatrix} -\boldsymbol{x}_1- & 0 \\ -\boldsymbol{w}- & 1 \\ -\boldsymbol{x}_3- & 0 \end{pmatrix} & \begin{pmatrix} -\boldsymbol{x}_1- & 0 \\ -\tilde{\boldsymbol{x}}_2- & 0 \\ -\boldsymbol{x}_3- & 0 \end{pmatrix} \\
\Psi_{\boldsymbol{X}}(\boldsymbol{Z}_0)=0, \Psi_{\tilde{\boldsymbol{X}}}(\boldsymbol{Z}_0)=0 & \Psi_{\boldsymbol{X}}(\boldsymbol{Z}_1)>0, \Psi_{\tilde{\boldsymbol{X}}}(\boldsymbol{Z}_1)=0 & \Psi_{\boldsymbol{X}}(\boldsymbol{Z}_2)>0, \Psi_{\tilde{\boldsymbol{X}}}(\boldsymbol{Z}_2)>0 & \Psi_{\boldsymbol{X}}(\boldsymbol{Z}_3)=0, \Psi_{\tilde{\boldsymbol{X}}}(\boldsymbol{Z}_3)>0
\end{array}
$$

Note that we can sample the row $\boldsymbol{w}$ by adding noise to the rows $\boldsymbol{x}_2$ or $\tilde{\boldsymbol{x}}_2$ only if the upper-level smoothing distribution $\phi$ first selects the perturbed row ($\boldsymbol{\tau}_2 = 1$). In general, we can only sample the same matrix $\boldsymbol{Z}$ from both distributions $\Psi_{\boldsymbol{X}}$ and $\Psi_{\tilde{\boldsymbol{X}}}$ if the upper-level smoothing distribution first selects all perturbed rows $i \in \mathcal{C}$. We group all those $\boldsymbol{Z}$ in the region $\mathcal{R}_2$.

**Partitioning into regions.** By appending the indicator $\boldsymbol{\tau}$ to the data we obtain four disjoint regions. To formally characterize these regions, we define the reachable set as $\mathcal{A} \triangleq \mathcal{A}_1 \cup \mathcal{A}_2 \subseteq \mathcal{Z}^{N \times (D+1)}$ with $\mathcal{A}_1 \triangleq \{\boldsymbol{Z} \mid \forall i : (\boldsymbol{\tau}_i = 0) \Rightarrow (\boldsymbol{z}_i = \boldsymbol{x}_i|0)\}$, and $\mathcal{A}_2 \triangleq \{\boldsymbol{Z} \mid \forall i : (\boldsymbol{\tau}_i = 0) \Rightarrow (\boldsymbol{z}_i = \tilde{\boldsymbol{x}}_i|0)\}$, where $\boldsymbol{x}_i|0$ denotes concatenation. Intuitively, $\mathcal{A}$ only contains matrices that can be sampled from $\Psi$. Now we can define the regions and derive the required probabilities to sample $\boldsymbol{Z}$ of each region:

**Proposition 1.** *The regions $\mathcal{R}_0, \mathcal{R}_1, \mathcal{R}_2$ and $\mathcal{R}_3$ are disjoint and partition the space $\mathcal{Z}^{N \times (D+1)}$:*

- $\mathcal{R}_0 \triangleq \mathcal{Z}^{N \times (D+1)} \setminus \mathcal{A}$
- $\mathcal{R}_1 \triangleq \{\boldsymbol{Z} \in \mathcal{A} \mid \exists i \in \mathcal{C} : \boldsymbol{\tau}_i = 0, \boldsymbol{z}_i = \boldsymbol{x}_i|0\}$
- $\mathcal{R}_2 \triangleq \{\boldsymbol{Z} \in \mathcal{A} \mid \forall i \in \mathcal{C} : \boldsymbol{\tau}_i = 1\}$
- $\mathcal{R}_3 \triangleq \{\boldsymbol{Z} \in \mathcal{A} \mid \exists i \in \mathcal{C} : \boldsymbol{\tau}_i = 0, \boldsymbol{z}_i = \tilde{\boldsymbol{x}}_i|0\}$

**Proposition 2.** *Given $\Psi_{\boldsymbol{X}}$ and $\Psi_{\tilde{\boldsymbol{X}}}$, the regions of Proposition 1 and $\Delta \triangleq 1 - p^{|\mathcal{C}|}$ we have*

$$
\Pr_{\boldsymbol{Z} \sim \Psi_{\boldsymbol{X}}}[\boldsymbol{Z} \in \mathcal{R}_1] = \Delta \qquad \Pr_{\boldsymbol{Z} \sim \Psi_{\boldsymbol{X}}}[\boldsymbol{Z} \in \mathcal{R}_2] = 1 - \Delta \qquad \Pr_{\boldsymbol{Z} \sim \Psi_{\boldsymbol{X}}}[\boldsymbol{Z} \in \mathcal{R}_3] = 0
$$

$$
\Pr_{\boldsymbol{Z} \sim \Psi_{\tilde{\boldsymbol{X}}}}[\boldsymbol{Z} \in \mathcal{R}_1] = 0 \qquad \Pr_{\boldsymbol{Z} \sim \Psi_{\tilde{\boldsymbol{X}}}}[\boldsymbol{Z} \in \mathcal{R}_2] = 1 - \Delta \qquad \Pr_{\boldsymbol{Z} \sim \Psi_{\tilde{\boldsymbol{X}}}}[\boldsymbol{Z} \in \mathcal{R}_3] = \Delta
$$

Proofs in Appendix C. Intuitively, $1 - \Delta = p^{|\mathcal{C}|}$ is the probability that all perturbed rows $i \in \mathcal{C}$ are selected ($\boldsymbol{\tau}_i = 1$), and $\Delta$ that at least one row $i \in \mathcal{C}$ is not selected by $\phi$. Lastly, the probability to sample $\boldsymbol{Z} \in \mathcal{R}_0$ is just 0. We visualize Proposition 1 and Proposition 2 in Figure 3.

**Hierarchical smoothing certificates.** Having derived a partitioning of $\mathcal{Z}^{N\times(D+1)}$ as above, we can apply the NP-Lemma for discrete random variables (Lemma 2) on the upper-level distribution $\phi$. Notably, we show that the problem of certification under hierarchical smoothing can be reduced to proving robustness for the lower-level distribution $\mu$ on the adversarially perturbed rows $\mathcal{C}$ only:

**Theorem 1** (Neyman-Pearson lower bound for hierarchical smoothing). *Given fixed $\boldsymbol{X}, \tilde{\boldsymbol{X}} \in \mathcal{X}^{N\times D}$, let $\mu_{\boldsymbol{X}}$ denote a smoothing distribution that operates independently on matrix elements, and $\Psi_{\boldsymbol{X}}$ the induced hierarchical distribution over $\mathcal{Z}^{N\times(D+1)}$. Given label $y \in \mathcal{Y}$ and the probability $p_{\boldsymbol{X},y}$ to classify $\boldsymbol{X}$ as $y$ under $\Psi$. Define $\hat{S}_\kappa \triangleq \left\{ \boldsymbol{W} \in \mathcal{X}^{|\mathcal{C}|\times D} : \mu_{\tilde{\boldsymbol{X}}_\mathcal{C}}(\boldsymbol{W}) \leq \kappa \cdot \mu_{\boldsymbol{X}_\mathcal{C}}(\boldsymbol{W}) \right\}$. We have:*

$$\underline{p_{\tilde{\boldsymbol{X}},y}} = \Pr_{\boldsymbol{W}\sim\mu_{\tilde{\boldsymbol{X}}_\mathcal{C}}}[\boldsymbol{W} \in \hat{S}_\kappa] \cdot (1-\Delta) \quad \text{with } \kappa \in \mathbb{R}_+ \text{ s.t.} \quad \Pr_{\boldsymbol{W}\sim\mu_{\boldsymbol{X}_\mathcal{C}}}[\boldsymbol{W} \in \hat{S}_\kappa] = \frac{p_{\boldsymbol{X},y}-\Delta}{1-\Delta}$$

*where $\boldsymbol{X}_\mathcal{C}$ and $\tilde{\boldsymbol{X}}_\mathcal{C}$ denote those rows $\boldsymbol{x}_i$ of $\boldsymbol{X}$ and $\tilde{\boldsymbol{x}}_i$ of $\tilde{\boldsymbol{X}}$ with $i \in \mathcal{C}$, that is $\boldsymbol{x}_i \neq \tilde{\boldsymbol{x}}_i$.*

*Proof sketch* (Full proof in Appendix C). In the worst-case all matrices in $\mathcal{R}_1$ are correctly classified. Note that this is the worst case since we cannot obtain any matrix of region $\mathcal{R}_1$ by sampling from the distribution $\Psi_{\tilde{\boldsymbol{X}}}$. Therefore the worst-case classifier first uses the budget $p_{\boldsymbol{X},y}$ on region $\mathcal{R}_1$ and we can subtract the probability $\Pr_{\boldsymbol{Z}\sim\Psi_{\boldsymbol{X}}}[\boldsymbol{Z} \in \mathcal{R}_1] = \Delta$ from the label probability $p_{\boldsymbol{X},y}$.

Since we never sample matrices of region $\mathcal{R}_3$ from the distribution $\Psi_{\boldsymbol{X}}$, the remaining correctly classified matrices must be in region $\mathcal{R}_2$, which one reaches with probability $1 - \Delta$ from both distributions $\Psi_{\boldsymbol{X}}$ and $\Psi_{\tilde{\boldsymbol{X}}}$. In region $\mathcal{R}_2$, however, we can simplify the problem to computing the certificate for $\mu$ on the perturbed rows to distribute the remaining probability. Since one never samples matrices of region $\mathcal{R}_0$ we do not have to consider it. The remaining statement follows from the Neyman-Pearson-Lemma (Neyman and Pearson, 1933; Tocher, 1950). $\qquad\square$

Note that we also derive the counterpart for discrete lower-level $\mu$ in Appendix F.

**Implications.** Notably, Theorem 1 implies that we can delegate robustness guarantees to the lower-level smoothing distribution $\mu$, compute the optimal value of Equation 1 under $\mu$ given the transformed probability $(p_{\boldsymbol{X},y}-\Delta)/(1-\Delta)$ and multiply the result with $(1-\Delta)$. This way we obtain the optimal value of Equation 1 under the hierarchical smoothing distribution $\Psi$ on the extended matrix space and thus robustness guarantees for hierarchical smoothing. This means our certificates are highly flexible and can integrate the whole suite of existing smoothing distributions with independent noise per dimension (Lecuyer et al., 2019; Cohen et al., 2019; Lee et al., 2019; Yang et al., 2020a). This is in contrast to all existing approaches where one has to come up with novel smoothing distributions and derive certificates from scratch once new threat models are introduced.

**Special cases.** We discuss two special cases of the probability $p$ for smoothing rows: First, if the probability to select rows for smoothing is $p = 1$ (intuitively, we always smooth the entire matrix), then we have $\Delta = 1 - p^{|\mathcal{C}|} = 0$ for any number of perturbed rows $|\mathcal{C}|$ and with Theorem 1 we get $\underline{p_{\tilde{\boldsymbol{X}},y}}(\Psi) = \underline{p_{\tilde{\boldsymbol{X}},y}}(\mu)$. That is in this special case we obtain the original certificate for the lower-level smoothing distribution $\mu$. Note that in this case the certificate ignores $r$. Second, if the probability to select rows is $p = 0$ (intuitively, we never sample smoothed matrices), we have $\Delta = 1 - p^{|\mathcal{C}|} = 1$ for any $|\mathcal{C}| \geq 1$, and with Theorem 1 we get $\underline{p_{\tilde{\boldsymbol{X}},y}} = 0$, that is we do not obtain any certificates.

## 5.2 Regional robustness certificates for hierarchical randomized smoothing

So far we can only guarantee that the prediction is robust for a specific perturbed input $\tilde{\boldsymbol{X}} \in \mathcal{B}^r_{p,\epsilon}(\boldsymbol{X})$. To ensure that the smoothed classifier is robust to the entire treat model $\mathcal{B}^r_{p,\epsilon}(\boldsymbol{X})$ we have to guarantee $\min_{\tilde{\boldsymbol{X}}\in\mathcal{B}^r_{p,\epsilon}(\mathbf{X})} \underline{p_{\tilde{\boldsymbol{X}},y^*}} > 0.5$. In the following we show that it is sufficient to compute Theorem 1 for $\Delta = 1 - p^r$ with the largest radius $r$. In fact, the final certificate is independent of which rows are perturbed, as long the two matrices differ in exactly $r$ rows the certificate is the same.

**Proposition 3.** *Given clean $\boldsymbol{X} \in \mathcal{X}^{N\times D}$. Consider the set $\mathcal{Q}^r_{p,\epsilon}(\boldsymbol{X})$ of inputs at a fixed distance $r$ (i.e. $|\mathcal{C}| = r$ for all $\tilde{\boldsymbol{X}} \in \mathcal{Q}^r_{p,\epsilon}(\boldsymbol{X})$). Then we have $\min_{\tilde{\boldsymbol{X}}\in\mathcal{B}^r_{p,\epsilon}(\mathbf{X})} \underline{p_{\tilde{\boldsymbol{X}},y}} = \min_{\tilde{\boldsymbol{X}}\in\mathcal{Q}^r_{p,\epsilon}(\mathbf{X})} \underline{p_{\tilde{\boldsymbol{X}},y}}$.*

*Proof.* The probability $1 - \Delta = p^{|\mathcal{C}|}$ is monotonously decreasing in the number of perturbed rows $|\mathcal{C}|$. Thus the lower bound $\underline{p_{\tilde{\boldsymbol{X}},y}}$ is also monotonously decreasing in $|\mathcal{C}|$ (see Theorem 1). It follows for fixed $\epsilon$ that $\underline{p_{\tilde{\boldsymbol{X}},y}}$ is minimal at $|\mathcal{C}| = r$, i.e. the largest possible radius under our threat model $\mathcal{B}^r_{p,\epsilon}(\boldsymbol{X})$.

# 6 Initializing hierarchical randomized smoothing

In the previous section we derive robustness certificates for hierarchical randomized smoothing for any lower-level smoothing distribution $\mu$ by deriving a general lower bound on the probability $p_{\tilde{\boldsymbol{X}},y}$ in Theorem 1. In this section we instantiate our hierarchical smoothing framework with specific lower-level smoothing distributions for discrete and continuous data.

## 6.1 Hierarchical randomized smoothing using Gaussian isotropic smoothing

Concerning continuous data $\mathcal{X} = \mathbb{R}$, consider the threat model $\mathcal{B}_{p,\epsilon}^r(\boldsymbol{X})$ for $p = 2$. We initialize the lower-level smoothing distribution of hierarchical smoothing with the Gaussian distribution (Cohen et al., 2019) (as introduced in Section 3), since it is specifically designed and tight for the well-studied $\ell_2$-norm threat model. For our purposes we apply the Gaussian noise independently on the matrix entries. In the following we present the binary-class certificates for hierarchical randomized smoothing induced by isotropic Gaussian smoothing:

**Corollary 1.** *Given continuous $\mathcal{X} \triangleq \mathbb{R}$, consider the threat model $\mathcal{B}_{2,\epsilon}^r(\boldsymbol{X})$ for fixed $\boldsymbol{X} \in \mathcal{X}^{N \times D}$. Initialize the hierarchical smoothing distribution $\Psi$ using the isotropic Gaussian distribution $\mu_{\boldsymbol{X}}(\boldsymbol{W}) \triangleq \prod_{i=1}^N \mathcal{N}(\boldsymbol{w}_i | \boldsymbol{x}_i, \sigma^2 \boldsymbol{I})$ that applies noise independently on each matrix element. Let $y^* \in \mathcal{Y}$ denote the majority class and $p_{\boldsymbol{X},y^*}$ the probability to classify $\boldsymbol{X}$ as $y^*$ under $\Psi$. Then the smoothed classifier $g$ is certifiably robust $g(\boldsymbol{X}) = g(\tilde{\boldsymbol{X}})$ for any $\tilde{\boldsymbol{X}} \in \mathcal{B}_{2,\epsilon}^r(\boldsymbol{X})$ if*

$$\epsilon < \sigma \left( \Phi^{-1} \left( \frac{p_{\boldsymbol{X},y^*} - \Delta}{1 - \Delta} \right) - \Phi^{-1} \left( \frac{1}{2(1 - \Delta)} \right) \right)$$

*where $\Phi^{-1}$ denotes the inverse CDF of the normal distribution and $\Delta \triangleq 1 - p^r$.*

Proof in Appendix E. We also derive the corresponding multi-class certificates in Appendix E.

## 6.2 Hierarchical randomized smoothing using sparse smoothing

To demonstrate our certificates in discrete domains we consider binary data $\mathcal{X} = \{0, 1\}$ and model adversaries that delete $r_d$ ones (flip $1 \rightarrow 0$) and add $r_a$ ones (flip $0 \rightarrow 1$), that is we consider the ball $\mathcal{B}_{r_a,r_d}^r(\boldsymbol{X})$ (see Appendix B for a formal introduction). We initialize the lower-level smoothing distribution of hierarchical smoothing with the sparse smoothing distribution proposed by Bojchevski et al. (2020): $p(\mu(\boldsymbol{x})_i \neq \boldsymbol{x}_i) = p_+^{1-\boldsymbol{x}_i} p_-^{\boldsymbol{x}_i}$, introducing two different noise probabilities to flip $0 \rightarrow 1$ with probability $p_+$ and $1 \rightarrow 0$ with probability $p_-$. The main idea is that the different flipping probabilities allow to preserve sparsity of the underlying data, making this approach particularly useful for graph-structured data. Note that we can consider the discrete certificate of Lee et al. (2019) as a special case (Bojchevski et al., 2020). We derive the corresponding certificates in Appendix F.

## 6.3 Hierarchical randomized smoothing using ablation smoothing

Lastly, randomized ablation (Levine and Feizi, 2020b) is a smoothing distribution where the input is not randomly smoothed but masked, e.g. by replacing the input with a special ablation token $\boldsymbol{t} \notin \mathcal{X}^D$ that does not exist in the original space. There are different variations of ablation and we carefully discuss such differences in Appendix G. By choosing a smoothing distribution $\mu$ that ablates all selected rows we can prove the following connection between hierarchical and ablation smoothing:

**Corollary 2.** *Initialize the hierarchical smoothing distribution $\Psi$ with the ablation smoothing distribution $\mu_{\boldsymbol{x}}(\boldsymbol{t}) = 1$ for ablation token $\boldsymbol{t} \notin \mathcal{X}^D$ and $\mu_{\boldsymbol{x}}(\boldsymbol{w}) = 0$ for $\boldsymbol{w} \in \mathcal{X}^D$ otherwise (i.e. we always ablate selected rows). Define $\Delta \triangleq 1 - p^r$. Then the smoothed classifier $g$ is certifiably robust $g(\boldsymbol{X}) = g(\tilde{\boldsymbol{X}})$ for any $\tilde{\boldsymbol{X}} \in \mathcal{B}_{2,\epsilon}^r(\boldsymbol{X})$ if $p_{\boldsymbol{X},y} - \Delta > 0.5$.*

Proof in Appendix G. Interestingly, Corollary 1 and Corollary 2 show that hierarchical smoothing is a generalization of additive noise and ablation certificates. The additional column indicates which rows will be ablated, but instead of ablating we only add noise to them. In contrast to ablation smoothing (which just checks if $p_{\boldsymbol{X},y} - \Delta > 0.5$) we further "utilize" the budget $p_{\boldsymbol{X},y} - \Delta$ by plugging it into the lower-level certificate. Notably, we are the first to generalize beyond ablation and additive noise certificates, and our certificates are in fact orthogonal to all existing robustness certificates that are based on randomized smoothing.

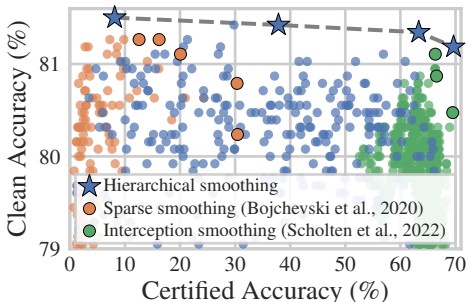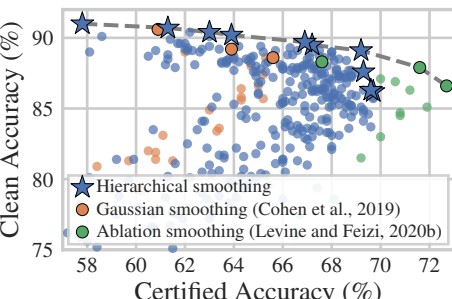

Figure 4: Hierarchical smoothing significantly expands the Pareto-front w.r.t. robustness and accuracy in node and image classification. Left: Discrete hierarchical smoothing for node classification, smoothed GAT on Cora-ML ($r = 1, r_d = 40, r_a = 0$). Right: Continuous hierarchical smoothing for image classification, smoothed ResNet50 on CIFAR10 ($r = 3, \epsilon = 0.35$). Non-smoothed GAT achieves $80\% \pm 2\%$ clean accuracy on Cora-ML, ResNet50 $94\%$ on CIFAR10. Large circles and stars are dominating points for each certificate. Dashed lines connect dominating points across methods.

# 7 Experimental evaluation

In this section we highlight the importance of hierarchical smoothing in image and node classification, instantiating our framework with two well-established smoothing distributions for continuous (Cohen et al., 2019) and discrete data (Bojchevski et al., 2020). In the following we present experiments supporting our main findings. We refer to Appendix A for more experimental results and to Appendix B for the full experimental setup and instructions to reproduce our results.

**Threat models.** For images we model adversaries that perturb at most $r$ pixels of an image, where the perturbation over all channels is bounded by $\epsilon$ under the $\ell_2$-norm. For graphs we model adversaries that perturb binary features of at most $r$ nodes by inserting at most $r_a$ and deleting at most $r_d$ ones.

**Datasets and models.** For image classification we train ResNet50 (He et al., 2016) on CIFAR10 (Krizhevsky et al., 2009) consisting of images with three channels of size 32x32 categorized into 10 classes. We certify the models on a random but fixed subset of 1,000 test images. For node classification we train graph attention networks (GATs) (Velickovic et al., 2018) with two layers on Cora-ML (McCallum et al., 2000; Bojchevski and Günnemann, 2018) consisting of 2,810 nodes with 2,879 *binary* features, 7,981 edges and 7 classes. We train and certify GNNs in an inductive learning setting (Scholten et al., 2022). We defer results for more models and datasets to Appendix A.

**Experimental setup.** During training, we sample one smoothed matrix each epoch to train our models on smoothed data. Note that for hierarchical smoothing we also train our models on the higher-dimensional matrices. At test time, we use significance level $\alpha = 0.01$, $n_0 = 1,000$ samples for estimating the majority class and $n_1 = 10,000$ samples for certification. We report the classification accuracy of the smoothed classifier on the test set (*clean accuracy*), and the *certified accuracy*, that is the number of test samples that are correctly classified *and* certifiably robust for a given radius. For node classification we report results averaged over five random graph splits and model initializations.

**Baseline certificates.** We compare hierarchical smoothing to additive noise and ablation certificates since our method generalizes both into a novel framework. To this end, we implement the two additive noise certificates of Cohen et al. (2019) and Bojchevski et al. (2020) for continuous and discrete data, respectively. Concerning ablation certificates, we implement the certificate of Levine and Feizi (2020b) for image classification, which ablates entire pixels. For node classification we implement the generalization to GNNs of Scholten et al. (2022) that ablates all node features of entire nodes. We conduct exhaustive experiments to explore the space of smoothing parameters for each method (see details in Appendix B). To compare the three different approaches we fix the threat model and investigate the robustness-accuracy trade-off by comparing certified accuracy against clean accuracy.

**Hierarchical smoothing significantly expands the Pareto-front.** Notably, hierarchical smoothing is clearly expanding the Pareto-front when optimization for both – certified accuracy and clean accuracy (see Figure 4). Especially when the number of adversarial nodes or pixels is small but the feature perturbation is large, hierarchical smoothing is significantly dominating both baselines.

Interestingly, both baselines either sacrifice robustness over accuracy (or vice versa): While additive noise certificates obtain higher clean accuracy at worse certified accuracy, ablation certificates obtain strong certified accuracy at lower clean accuracy. In contrast, hierarchical smoothing allows to explore the entire space and significantly expands the Pareto-front of the robustness-accuracy trade-off under our threat model. This demonstrates that hierarchical smoothing is a more flexible framework and represents a useful and novel tool for analyzing the robustness of machine learning models in general.

**Entity-selection probability.** With hierarchical smoothing we introduce the entity-selection probability $p$ that allows to better control the robustness-accuracy trade-off under our threat model. Specifically, for larger $p$ we add more noise and increase robustness but also decrease accuracy. As usual in randomized smoothing, the optimal $p$ needs to be fine-tuned against a task, dataset and radius $r$. In our settings we found dominating points e.g. for $p = 0.81$ (Cora-ML) and $p = 0.85$ (CIFAR10).

## 8    Discussion

Our hierarchical smoothing approach overcomes limitations of randomized smoothing certificates: Instead of having to derive Lemma 1 from scratch for each smoothing distribution, we can easily integrate the whole suite of existing and future randomized smoothing certificates. Our framework is highly flexible and allows to certify robustness against a wide range of adversarial perturbations.

Hierarchical smoothing also comes with limitations: First, we inherit the limitations of ablation certificates in which the certifiable radius $r$ is bounded by the smoothing parameters independently of the classifier (Scholten et al., 2022). The underlying reason is that we do not evaluate the smoothed classifier on the entire space $\mathcal{A}$ since we cannot reach certain matrices as exploited in Section 5. Second, the classifier defined on the original matrix space is invariant with respect to the new dimension introduced by our smoothing distribution and not incorporating such invariances into the Neyman-Pearson-Lemma yields strictly looser guarantees (Schuchardt and Günnemann, 2022).

Beyond that, our certificates are highly efficient: The only additional cost for computing hierarchical smoothing certificates is to evaluate the algebraic term $\Delta = 1 - p^r$, which takes constant time (see also Appendix H). Since we train our classifiers on the extended matrices, we also allow classifiers to distinguish whether entities have been smoothed by the lower-level distribution (see Appendix A).

**Future work.** Future work can build upon our work towards even stronger certificates in theory and practice, for example by deriving certificates that are tight under the proposed threat model and efficient to compute at the same time. Future work can further (1) implement certificates for other $\ell_p$-norms (Levine and Feizi, 2021; Vorácek and Hein, 2023) and domains, (2) improve and assess adversarial robustness, and (3) introduce novel architectures and training techniques.

**Broader impact.** Our hierarchical smoothing certificates are highly flexible and provide provable guarantees for arbitrary ($\ell_p$-norm) threat models, provided that certificates for the corresponding lower-level smoothing distribution exist. Therefore our contributions impact the certifiable robustness of a large variety of models in discrete and continuous domains and therefore the field of reliable machine learning in general: Our robustness certificates provide novel ways of assessing, understanding and improving the adversarial robustness of machine learning models.

## 9    Conclusion

With hierarchical smoothing we propose the first hierarchical (mixture) smoothing distribution for complex data where adversaries can perturb only a subset of all entities (e.g. pixels in an image, or nodes in a graph). Our main idea is to add noise in a more targeted manner: We first select a subset of all entities and then we add noise to them. By certifying robustness under this hierarchical smoothing distribution we achieve stronger robustness guarantees while still maintaining high accuracy. Overall, our certificates for hierarchical smoothing represent novel, flexible tools for certifying the adversarial robustness of machine learning models towards more reliable machine learning.

## Acknowledgments and Disclosure of Funding

This work has been funded by the Munich Center for Machine Learning, by the DAAD program Konrad Zuse Schools of Excellence in Artificial Intelligence (sponsored by the Federal Ministry of Education and Research), and by the German Research Foundation, grant GU 1409/4-1. The authors of this work take full responsibility for its content.

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

# A Additional experiments and results

In the following we provide additional results for our experiments with hierarchical randomized smoothing for the tasks of node and image classification.

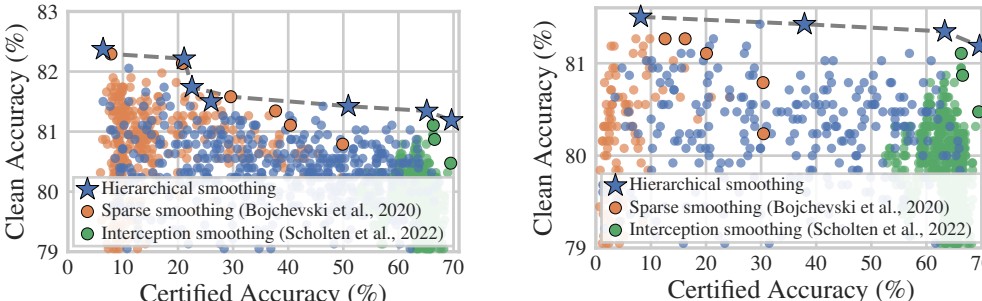

Figure 5: Discrete hierarchical smoothing significantly extends the Pareto-front w.r.t. robustness-accuracy (smoothed GAT on Cora-ML). Left: $r=1, r_a=0, r_d=20$. Right: $r=1, r_a=0, r_d=40$.

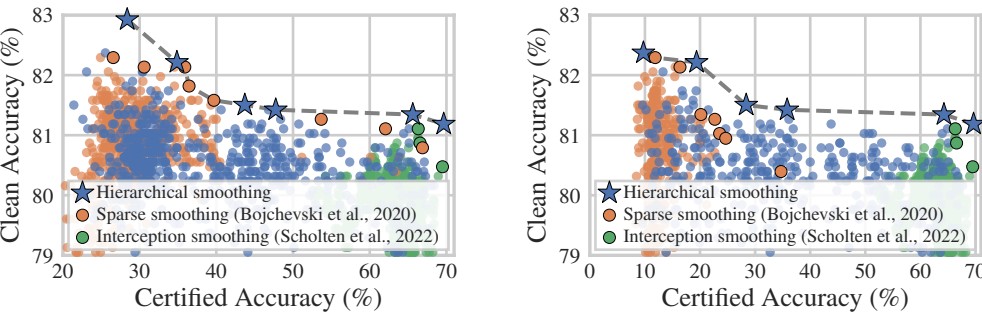

Figure 6: Discrete hierarchical smoothing significantly extends the Pareto-front w.r.t. robustness-accuracy (smoothed GAT on Cora-ML). Left: $r=1, r_a=5, r_d=0$. Right: $r=1, r_a=10, r_d=0$.

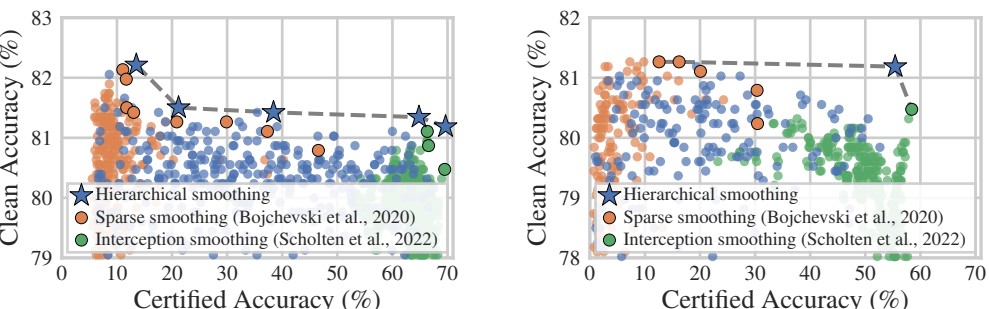

Figure 7: Discrete hierarchical smoothing significantly extends the Pareto-front w.r.t. robustness-accuracy (smoothed GAT on Cora-ML). Left: $r=1, r_a=5, r_d=15$. Right: $r=2, r_a=0, r_d=40$.

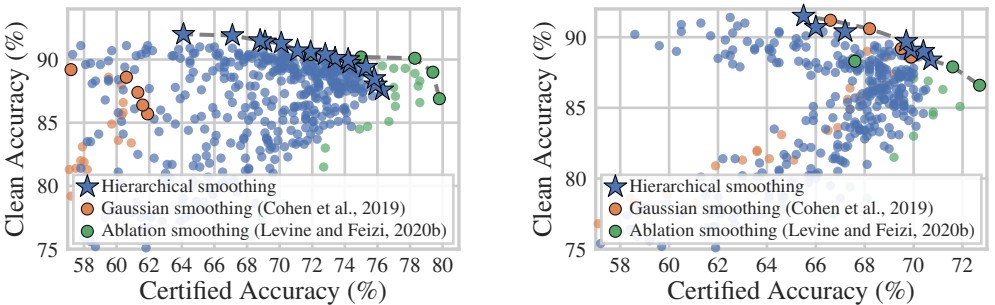

Figure 8: Continuous hierarchical smoothing significantly extends the Pareto-front w.r.t. robustness-accuracy (smoothed ResNet50 on CIFAR10). Left: $r=2, \epsilon=0.4$. Right: $r=3, \epsilon=0.3$.

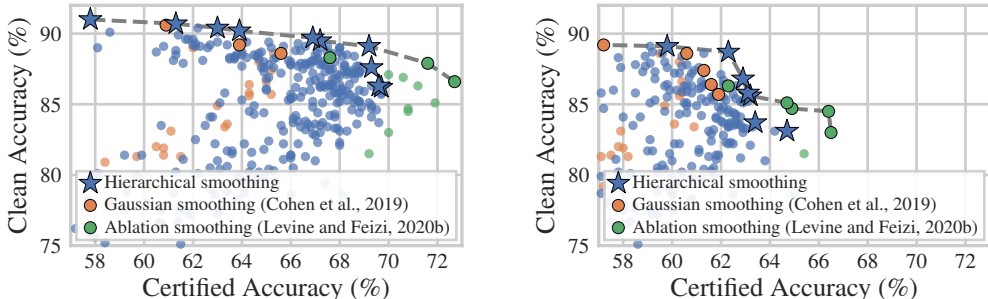

Figure 9: Continuous hierarchical smoothing significantly extends the Pareto-front w.r.t. robustness-accuracy (smoothed ResNet50 on CIFAR10). Left: $r = 3, \epsilon = 0.35$. Right: $r = 4, \epsilon = 0.4$.

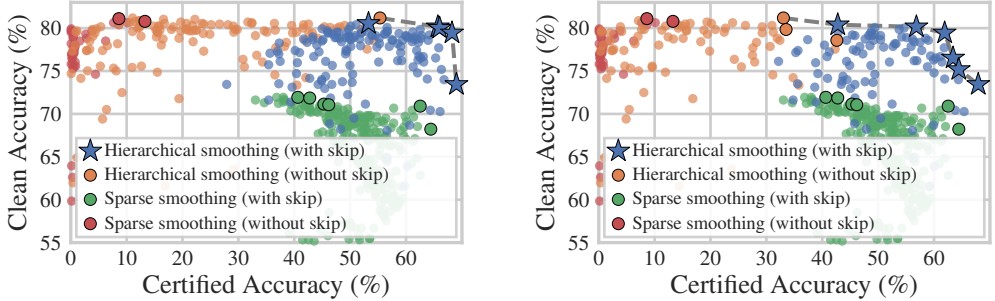

Figure 10: Skip-connections for hierarchical and sparse smoothing significantly extend the Pareto-front (smoothed GAT on Cora-ML). Left: $r = 2, r_a = 0, r_d = 70$. Right: $r = 3, r_a = 0, r_d = 70$.

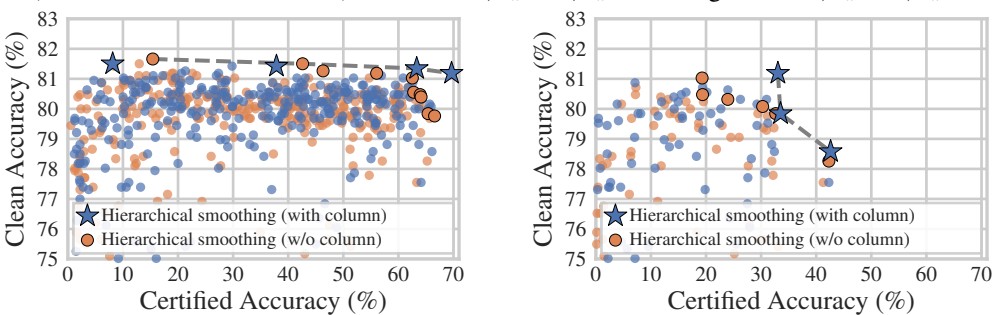

Figure 11: Hierarchical smoothing with and without appended column (smoothed GAT on Cora-ML). Left: $r = 1, r_a = 0, r_d = 40$. Right: $r = 3, r_a = 0, r_d = 70$.

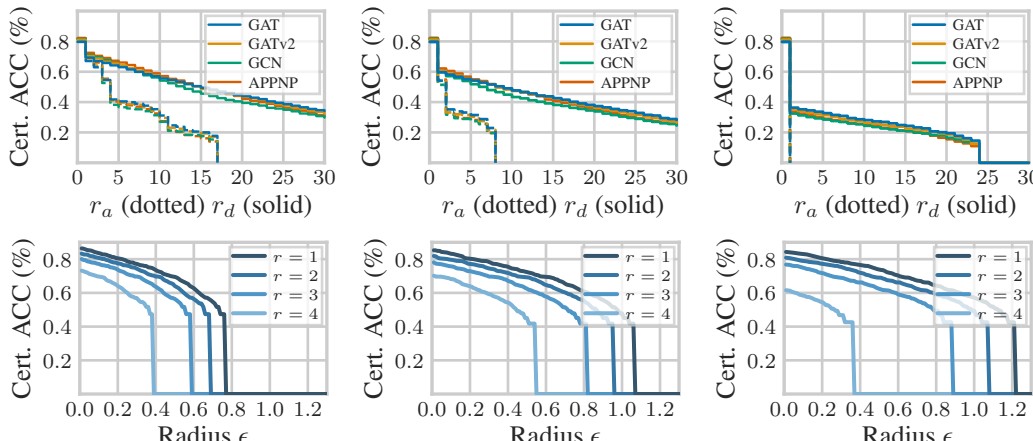

Figure 12: Upper row: Smoothed GNNs on Cora-ML ($p = 0.8, p_a = 0.006, p_d = 0.88$) for $r = 1, 2, 3$ from left to right. Lower row: Smoothed ResNet50 on CIFAR10 for different radii $r$ (Left: $k = 150, \sigma = 0.25$. Middle: $k = 150, \sigma = 0.35$. Right: $k = 160, \sigma = 0.4$.)

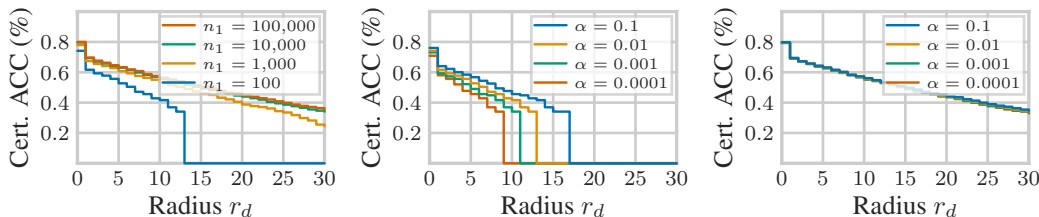

Figure 13: Smoothed GAT models on CoraML ($r = 1, r_a = 0$). Different number of Monte-Carlo samples and the effect on the certifiable robustness. Left: $\alpha = 0.01$. Middle: $n_1 = 100$. Right: $n_1 = 10{,}000$. Different clean accuracies at $r_d = 0$ are due to abstained predictions.

For node classification we experimentally observe that smoothed GAT models are more robust on average (Figure 12 upper row). We therefore decide to show more results for smoothed GAT models in the following. Recall that $r$ is the number of adversarial nodes, $r_d$ is the number of adversarial deletions ($1 \rightarrow 0$), and $r_a$ the number of adversarial insertions ($0 \rightarrow 1$). In Figure 5 we show results for varying $r_d$ with fixed $r = 1$ and $r_a = 0$. In Figure 6 we further vary $r_a$ with fixed $r = 1$ and $r_d = 0$. In Figure 7 we show results for $r = 1$, $r_d > 0$ and $r_a > 0$ and separately results for $r = 2$. For image classification, recall that $r$ is the number of pixels controlled by the adversary and $\epsilon$ the perturbation strength in $\ell_2$-distance across channels. In Figure 8 and Figure 9 we provide additional results for different radii $r$ and $\epsilon$.

Clearly, hierarchical randomized smoothing is significantly expanding the Pareto-front w.r.t. the robustness-accuracy trade-off in all of these settings. Since our certificates generalize ablation and additive noise certificates into a new common framework, our method performs either better or on-par with the existing methods under our threat model. Note that in special threat models where e.g. adversaries have access to all nodes ($r = N$), our certificates correspond to standard randomized smoothing and consequently perform on-par with existing methods.

In Figure 12 (upper row) we compare different GNN architectures for the task of node classification and generally observe that smoothed GAT models are more robust on average. In Figure 12 (lower row) we provide further results for different smoothing parameters when certifying the robustness of smoothed ResNet50 models on CIFAR10. Note the different effects on the certifiable robustness at different radii $r$ when increasing $\sigma$ and $k$ (increasing $k$ corresponds to decreasing $p$).

In Figure 10 we study the setting where adversaries do not have access to target nodes in the graph but only to nodes in the first- and second-hop neighborhoods (this setting has been previously studied by Scholten et al. (2022)). Such threat models allow us to introduce skip-connections: We can forward the (unsmoothed) features of target nodes separately through the GNN (without graph structure) and add the final representations on the representations obtained with smoothing. In Figure 10 we experimentally observe that such skip-connections can significantly expand the Pareto-front w.r.t. robustness and accuracy of sparse and hierarchical smoothing.

In Figure 11 we perform experiments with and without the additional additional column for smoothed GAT models on CoraML. Notably, training the models on the extended matrices and using the additional column at inference can result in better classifiers: As we experimentally observe for graph-structured data, using the additional column significantly extends the Pareto-front w.r.t. robustness and accuracy under specific threat models.

In Figure 13 we study the effect of different numbers of Monte-Carlo samples on the certificate strength. We observe that for our setting using $\alpha = 0.01$ and $n_1 = 10{,}000$ is sufficient to compare our method to the baselines. Note that in practice one would have to fix $\alpha$ and $n_0, n_1$ prior to computing certificates to ensure their statistical validity.

**Error bars.** For the node classification task we run every experiment 5 times with different dataset splits and model initializations and report averaged results. The average standard deviation is (3%, 3%, 4%, 4%) in clean accuracy and (3%, 3%, 2%, 3%) in certified accuracy (across all node classification experiments, hierarchical smoothing only, sparse smoothing only, and ablation smoothing only, respectively).

# B Full experimental setup (Section 7)

We provide detailed information on the experimental setup as described in Section 7 to ensure reproducibility of our results. Note that we also upload our implementation.[4]

## B.1 Experimental setup for the task of node classification

**Datasets and models.** We implement smoothed classifiers for four architectures with two message-passing layers: Graph convolutional networks (GCN) (Kipf and Welling, 2017), graph attention networks (GAT and GATv2) (Velickovic et al., 2018; Brody et al., 2022), and APPNP (Klicpera et al., 2019). We train models on the citation dataset Cora-ML (Bojchevski and Günnemann, 2018; McCallum et al., 2000) consisting of 2,810 nodes with 2,879 *binary* features, 7,981 edges and 7 classes. We follow the standard procedure in graph machine learning (Shchur et al., 2018) and preprocess graphs into undirected graphs containing only their largest connected component.

**Model details.** We implement all GNN models for two message-passing layers. We use 64 hidden channels for GCN (Kipf and Welling, 2017) and APPNP (Klicpera et al., 2019), and 8 attention heads for 8 hidden channels for GAT and GATv2 (Velickovic et al., 2018; Brody et al., 2022). For APPNP we further set the hyperparameter k_hops to $10$ and the teleport probability to $0.15$.

**Node classification task.** For GNNs we follow the setup in (Scholten et al., 2022). We evaluate our certificates for the task of node classification in semi-supervised *inductive* learning settings: We label $20$ randomly drawn nodes per class for training and validation, and 10% of the nodes for testing. We use the labelled training nodes and all remaining unlabeled nodes as training graph, and insert the validation and test nodes into the graph at validation and test time, respectively. We train our models on training nodes, tune them on validation nodes and compute certificates for all test nodes. Note that transductive settings come with shortcomings when evaluating robustness certificates (Scholten et al., 2022), or in adversarial robustness on graph-structured data in general (Gosch et al., 2023a).

**Training details.** We train models full-batch using Adam (learning rate $= 0.001$, $\beta_1 = 0.9$, $\beta_2 = 0.999$, $\epsilon = e{-}08$, weight decay $= 5e{-}04$) for a maximum of $1,000$ epochs with cross-entropy loss. Note that we implement early stopping after $50$ epochs. We also use a dropout of $0.5$ on the hidden node representations after the first graph convolution for GCN and GAT, and additionally a dropout of $0.5$ on the attention coefficients for GAT and GATv2.

**Training-time smoothing parameters.** During training we sample one smoothed feature matrix (i.e. a matrix with (partially) added noise) in each epoch and then we normally proceed training with the noised feature matrix. Unless stated differently we also train our models on the higher-dimensional matrices, i.e. we append the additional column to the matrices. Note that we use the same smoothing parameters during training as for certification.

**Experimental evaluation.** We compare hierarchical smoothing to the additive noise baseline by Bojchevski et al. (2020) and the ablation baseline Scholten et al. (2022) as follows: We run $1,000$ experiments for each method by (1) randomly drawing specific smoothing parameters ($p$ and $p_+, p_-$, depending on the method), (2) training $5$ smoothed classifiers with different seeds for the datasplit and model initialization, (3) computing robustness certificates under the corresponding smoothing distribution, and (4) averaging resulting clean and certified accuracies.

**Exploring the Pareto-front.** In our exhaustive experiments we randomly explore the entire space of possible parameter combinations. The reason for this is to demonstrate Pareto-optimality, i.e. to find all ranges of smoothing parameters for which we can offer both better robustness and accuracy. Note that in practical settings one would have to conduct significantly less experiments to find suitable parameters, and the task of efficiently finding the optimal parameters is a research direction orthogonal to deriving robustness certificates.

**Smoothing parameters for node classification.** We randomly sample from the following space of possible smoothing parameters (we draw from a uniform distribution over these intervals):

- Discrete hierarchical smoothing: $p \in [0.51, 0.993]$, $p_+ \in [0, 0.05]$, $p_- \in [0.5, 1]$,
- Sparse smoothing (additive noise baseline): $p_+ \in [0, 0.05]$, $p_- \in [0.5, 1]$
- Interception smoothing (ablation smoothing baseline): $p \in [0.51, 0.993]$

---

[4]Project page: `https://www.cs.cit.tum.de/daml/hierarchical-smoothing`

Note that the search space of hierarchical smoothing is the cross product of the search spaces of the two baselines (sparse smoothing (Bojchevski et al., 2020) and interception smoothing (Scholten et al., 2022)). Notably, we still find that hierarchical smoothing is superior despite running 1,000 experiments equally for all three methods (see Section 7). Note that we run 1,000 experiments equally for all methods to establish an equal chance for random outliers in accuracy. Also note that for clarity of the plots we only include points with certified accuracies of larger than 0%.

## B.2    Experimental setup for the task of image classification

**Dataset and model architecture.** For the task of image classification we follow the experimental setup in (Levine and Feizi, 2020b) and train smoothed ResNet50 (He et al., 2016) on CIFAR10 (Krizhevsky et al., 2009). CIFAR10 consists of 50,000 training images and 10,000 test images with 10 classes (6,000 images per class). The colour images are of size 32x32 and have 3 channels. Note that for runtime reasons we only certify a subset of the entire test set: For all experiments we report clean and certified accuracy for the same subset consisting of 1,000 randomly sampled test images.

**Training details.** We train the ResNet50 model with stochastic gradient descent (learning rate 0.01, momentum 0.9, weight decay 5e-4) for 400 epochs using a batch-size of 128. We also deploy a cosine learning rate scheduler (Loshchilov and Hutter, 2017). At inference we use a batch size of 300. We append the additional indicator channel (see Figure 3) during training and certification. We train the smoothed classifiers with the same noise level as during certification.

**Image preprocessing and smoothing.** We augment the training set with random horizontal flips and random crops (with a padding of 4). We normalize all images using the channel-wise dataset mean [0.4914, 0.4822, 0.4465] and standard deviation [0.2023, 0.1994, 0.2010]. For the ablation smoothing baseline we follow the implementation of Levine and Feizi (2020b) and first normalize the images before appending three more channels to the images. Specifically, for each channel $x$ we append the channel $1 - x$ (resulting in 6 channels in total). Finally, we set all channels of ablated pixels to 0. We train and certify the extended images. See Levine and Feizi (2020b) for a detailed setup of their method. For hierarchical smoothing and the Gaussian smoothing baseline (Cohen et al., 2019) we first add the Gaussian noise and then normalize the images. Note that for hierarchical smoothing we extend the images with the additional indicator channel (Figure 3), resulting in 4 channels in total.

**Smoothing parameters for image classification.** Note that the ablation baseline (Levine and Feizi, 2020b) draws $k$ pixels to ablate from a uniform distribution $\mathcal{U}(k, N)$ over all possible subsets with $k$ pixels of an image with $N$ total pixels. This is in contrast to our approach, which selects pixels to add noise to with a probability of $p$. For the comparison we choose the pixel selection probability $p = 1 - \frac{k}{N}$ for every $k$, where $N$ is the number of pixels in an image (i.e. we select $k$ pixels in expectation). This way, increasing $k$ corresponds to decreasing $p$ and vice versa. We run experiments for every 10th $k$ from 0 to 500. Note that for $k$ larger that 512 we cannot obtain certificates with either method since then $\Delta > 0.5$. For Gaussian smoothing we run experiments for every 100th $\sigma$ in the interval $[0, 1]$, and for hierarchical smoothing for every 20th $\sigma$ in $[0, 1]$. Overall, we run 50 experiments for the ablation baseline (Levine and Feizi, 2020b), 100 experiments for the Gaussian noise baseline (Cohen et al., 2019), and 1,000 experiments for hierarchical smoothing.

### B.3 Further experimental details

**Certification parameters.** All presented plots for graphs show results for the tighter multi-class certificates, for images we compute the binary-class certificates. Our certificates are probabilistic (Cohen et al., 2019), and we use Monte-Carlo samples to estimate the smoothed classifiers with Clopper-Pearson confidence intervals for $\alpha = 0.01$ (thus our certificates hold with high probability of 99%). We also apply Bonferroni correction to ensure that the bounds hold simultaneously with confidence level $1 - \alpha$. We follow the procedure of Cohen et al. (2019) and, if not stated differently, use $n_0 = 1,000$ Monte-Carlo samples for estimating the majority class at test time and $n_1 = 10,000$ samples for the certification at test time.

**Reproducibility.** To ensure reproducibility we further use random seeds for all random processes including for model training, for drawing smoothing parameters and for drawing Monte-Carlo samples. We publish the source code including reproducibility instructions and all random seeds.

**Computational resources.** All experiments were performed on a Xeon E5-2630 v4 CPU with a NVIDIA GTX 1080TI GPU. In the following we report experiment runtimes (note that for node classification we compute certificates for 5 different smoothed models in each experiment): Regarding node classification experiments, the average runtime of hierarchical smoothing for the $1,000$ experiments is 1.6 hours. For the additive noise baseline 1.5 hours, and for the ablation baseline 0.92 hours. For image classification we only train a single classifier. The overall training and certification process takes (on average) 9.9 hours for hierarchical smoothing, 8.6 hours for Gaussian smoothing, and 13.3 hours for ablation smoothing.

**Third-party assets.** For the sparse smoothing baseline we use the implementation of Bojchevski et al. (2020).[5] For the ablation smoothing baseline for GNNs we use the implementation of Scholten et al. (2022).[6] We implement all GNNs using PyTorch Geometric (Fey and Lenssen, 2019).[7] The graph datasets are publicly available and can be downloaded also e.g. using PyTorch Geometric. The CIFAR10 dataset is also publicly available, for example via the torchvision library.[8]

---

[5]`https://github.com/abojchevski/sparse_smoothing`
[6]`https://github.com/yascho/interception_smoothing`
[7]`https://pytorch-geometric.readthedocs.io`
[8]`https://pytorch.org/vision/stable/index.html`

# C Robustness certificates for hierarchical smoothing (Proofs of Section 5)

In the following we show the statements of Section 5 about hierarchical smoothing certificates and correctness of the following certification procedure:

---
**Algorithm 1:** Binary-class Robustness Certificates for Hierarchical Randomized Smoothing

---
**Input:** $f$, $\boldsymbol{X}$, $p$, Lower-level smoothing distribution $\mu$, Radius $(r, \epsilon)$, $n_0$, $n_1$, $1 - \alpha$
counts0 $\leftarrow$ SampleUnderNoise($f$, $\boldsymbol{X}$, $p$, $\mu$, $n_0$) ;
counts1 $\leftarrow$ SampleUnderNoise($f$, $\boldsymbol{X}$, $p$, $\mu$, $n_1$) ;
$y_A \leftarrow$ top index in counts0 ;
$\underline{p_A} \leftarrow$ ConfidenceBound(counts1[$y_A$], $n_1$, $1 - \alpha$) ;
$\Delta \leftarrow 1 - p^r$ ;
$\underline{\tilde{p}_A} \leftarrow$ LowerLevelWorstCaseLowerBound($\mu$, $\epsilon$, $\frac{\underline{p_A} - \Delta}{1 - \Delta}$) ;
**if** $\underline{\tilde{p}_A} \cdot (1 - \Delta) > \frac{1}{2}$ **then**
  | **Return:** $y_A$ certified
**end**
**Return:** ABSTAIN

---

Here, LowerLevelWorstCaseLowerBound($\mu$,$\epsilon$, $\underline{p_A}$) computes the existing certificate, specifically the smallest $\underline{p_{\tilde{\boldsymbol{X}}, y_A}}$ over the entire threat model, e.g. $\Phi\left(\Phi^{-1}(\underline{p_A}) - \frac{\epsilon}{\sigma}\right)$ for Gaussian smoothing. The remaining certification procedure is analogous to the certification in (Cohen et al., 2019).

## C.1 Proofs of Section 5

First, define the space $\mathcal{Z}^{N \times (D+1)} \triangleq \mathcal{X}^{N \times D} \times \{0, 1\}^N$ and recall the definition of the reachable set $\mathcal{A} \triangleq \mathcal{A}_1 \cup \mathcal{A}_2$ with

$$\mathcal{A}_1 \triangleq \{\boldsymbol{Z} \in \mathcal{Z}^{N \times (D+1)} \mid \forall i : (\boldsymbol{\tau}_i = 0) \Rightarrow (\boldsymbol{z}_i = \boldsymbol{x}_i | 0)\},$$

and

$$\mathcal{A}_2 \triangleq \{\boldsymbol{Z} \in \mathcal{Z}^{N \times (D+1)} \mid \forall i : (\boldsymbol{\tau}_i = 0) \Rightarrow (\boldsymbol{z}_i = \tilde{\boldsymbol{x}}_i | 0)\}.$$

Intuitively, $\mathcal{A}$ contains only those $\boldsymbol{Z}$ that can be sampled from either $\Psi_{\boldsymbol{X}}$ or $\Psi_{\tilde{\boldsymbol{X}}}$ (given the lower-level smoothing distribution has infinite support). Note that these two sets are not necessarily disjoint, so we cannot apply the Neyman-Pearson Lemma for discrete random variables (Tocher, 1950) on the upper-level smoothing distribution $\phi$ yet. The following Proposition therefore provides a partitioning of the space $\mathcal{Z}^{N \times (D+1)}$:

**Proposition 1.** *The regions $\mathcal{R}_0, \mathcal{R}_1, \mathcal{R}_2$ and $\mathcal{R}_3$ are disjoint and partition the space $\mathcal{Z}^{N \times (D+1)}$:*

- $\mathcal{R}_0 \triangleq \mathcal{Z}^{N \times (D+1)} \setminus \mathcal{A}$
- $\mathcal{R}_2 \triangleq \{\boldsymbol{Z} \in \mathcal{A} \mid \forall i \in \mathcal{C} : \boldsymbol{\tau}_i = 1\}$
- $\mathcal{R}_1 \triangleq \{\boldsymbol{Z} \in \mathcal{A} \mid \exists i \in \mathcal{C} : \boldsymbol{\tau}_i = 0, \boldsymbol{z}_i = \boldsymbol{x}_i | 0\}$
- $\mathcal{R}_3 \triangleq \{\boldsymbol{Z} \in \mathcal{A} \mid \exists i \in \mathcal{C} : \boldsymbol{\tau}_i = 0, \boldsymbol{z}_i = \tilde{\boldsymbol{x}}_i | 0\}$

*Proof.* We want to show that the four sets above (1) are pairwise disjoint, and (2) partition $\mathcal{Z}^{N \times (D+1)}$, that is $\mathcal{R}_0 \uplus \mathcal{R}_1 \uplus \mathcal{R}_2 \uplus \mathcal{R}_3 = \mathcal{Z}^{N \times (D+1)}$.

Clearly we have $\mathcal{Z}^{N \times (D+1)} = \mathcal{R}_0 \uplus \mathcal{A}$ due to the definition of the region $\mathcal{R}_0$. Thus we only have to show $\mathcal{R}_1 \uplus \mathcal{R}_2 \uplus \mathcal{R}_3 = \mathcal{A}$:

(1) We show $\mathcal{R}_1, \mathcal{R}_2, \mathcal{R}_3$ are pairwise disjoint:

> We have $\mathcal{R}_2 \cap \mathcal{R}_1 = \emptyset$ and $\mathcal{R}_2 \cap \mathcal{R}_3 = \emptyset$ since in region $\mathcal{R}_2$ we have $\boldsymbol{\tau}_i = 1$ for all $i \in \mathcal{C}$, which does not hold for $\mathcal{R}_2$ or $\mathcal{R}_3$.
> Moreover, $\mathcal{R}_1 \cap \mathcal{R}_3 = \emptyset$ because if $\boldsymbol{Z} \in \mathcal{R}_1$ then $\boldsymbol{Z} \notin \mathcal{R}_3$ (and vice versa) due the to definition of $\mathcal{A}$ and $\mathcal{R}_1 \subseteq \mathcal{A}_1$, $\mathcal{R}_3 \subseteq \mathcal{A}_3$.

(2) We show $\mathcal{R}_1 \cup \mathcal{R}_2 \cup \mathcal{R}_3 = \mathcal{A}$:

> "$\subseteq$": Clearly $\mathcal{R}_1 \subseteq \mathcal{A}, \mathcal{R}_2 \subseteq \mathcal{A}$, and $\mathcal{R}_3 \subseteq \mathcal{A}$ by definition of the regions.
> "$\supseteq$": Consider any $\boldsymbol{Z} \in \mathcal{A} = \mathcal{A}_1 \cup \mathcal{A}_2$. Either (1) all perturbed rows are selected and $\boldsymbol{Z} \in \mathcal{R}_2$, or (2) at least one perturbed row is not selected ($\exists i : \boldsymbol{\tau}_i = 0$), in which case we either have $\boldsymbol{Z} \in \mathcal{R}_1$ or $\boldsymbol{Z} \in \mathcal{R}_3$, depending on whether $\boldsymbol{Z} \in \mathcal{A}_1$, $\boldsymbol{Z} \in \mathcal{A}_3$, respectively.

$\square$

We further have to derive the probability to sample $\boldsymbol{Z}$ of each region from both $\Psi_{\boldsymbol{X}}$ and $\Psi_{\tilde{\boldsymbol{X}}}$ for applying the Neyman-Pearson Lemma (Tocher, 1950) for discrete random variables on the upper-level smoothing distribution $\phi$ that selects rows before adding noise.

**Proposition 2.** *Given $\Psi_{\boldsymbol{X}}$ and $\Psi_{\tilde{\boldsymbol{X}}}$, the regions of Proposition 1 and $\Delta \triangleq 1 - p^{|\mathcal{C}|}$ we have*

$$\Pr_{\boldsymbol{Z} \sim \Psi_{\boldsymbol{X}}}[\boldsymbol{Z} \in \mathcal{R}_1] = \Delta \qquad \Pr_{\boldsymbol{Z} \sim \Psi_{\boldsymbol{X}}}[\boldsymbol{Z} \in \mathcal{R}_2] = 1 - \Delta \qquad \Pr_{\boldsymbol{Z} \sim \Psi_{\boldsymbol{X}}}[\boldsymbol{Z} \in \mathcal{R}_3] = 0$$
$$\Pr_{\boldsymbol{Z} \sim \Psi_{\tilde{\boldsymbol{X}}}}[\boldsymbol{Z} \in \mathcal{R}_1] = 0 \qquad \Pr_{\boldsymbol{Z} \sim \Psi_{\tilde{\boldsymbol{X}}}}[\boldsymbol{Z} \in \mathcal{R}_2] = 1 - \Delta \qquad \Pr_{\boldsymbol{Z} \sim \Psi_{\tilde{\boldsymbol{X}}}}[\boldsymbol{Z} \in \mathcal{R}_3] = \Delta$$

*Proof.* The probability to select all $|\mathcal{C}|$ perturbed rows is $p^r$ since we sample each $\boldsymbol{\tau}_i$ independently from a Bernoulli: $\boldsymbol{\tau}_i \overset{iid}{\sim} \text{Ber}(p)$. Consequently the probability to not select at least one perturbed row is $1 - p^r = \Delta$. The remaining follows from the fact that matrices from region $\mathcal{R}_1$ cannot be sampled from $\Psi_{\tilde{\boldsymbol{X}}}$ and matrices from $\mathcal{R}_3$ cannot be sampled from $\Psi_{\boldsymbol{X}}$. $\qquad\square$

Note that we need region $\mathcal{R}_0$ just for a complete partitioning of $\mathcal{Z}^{N \times (D+1)}$. This region contains essentially all matrices we cannot sample neither from $\Psi_{\boldsymbol{X}}$ nor $\Psi_{\tilde{\boldsymbol{X}}}$. For example, matrices with a row that differs from the same row in $\boldsymbol{X}$ and $\tilde{\boldsymbol{X}}$ without a previous selection of that row cannot be sampled from the two distributions – simply because we first select rows and then we only add noise to the rows that we selected. Thus we have $\Pr_{\boldsymbol{Z} \sim \Psi_{\boldsymbol{X}}}[\boldsymbol{Z} \in \mathcal{R}_0] = 0$ and $\Pr_{\boldsymbol{Z} \sim \Psi_{\tilde{\boldsymbol{X}}}}[\boldsymbol{Z} \in \mathcal{R}_0] = 0$.

Before we derive the lower bound on $\underline{p_{\tilde{\boldsymbol{X}},y}}$ we first restate the Neyman-Pearson Lemma as presented in Section 3:

**Lemma 1** (Neyman-Pearson lower bound). *Given $\boldsymbol{X}, \tilde{\boldsymbol{X}} \in \mathcal{X}^{N \times D}$, distributions $\mu_{\boldsymbol{X}}, \mu_{\tilde{\boldsymbol{X}}}$, class label $y \in \mathcal{Y}$, probability $p_{\boldsymbol{X},y}$ and the set $S_\kappa \triangleq \{\boldsymbol{W} \in \mathcal{X}^{N \times D} : \mu_{\tilde{\boldsymbol{X}}}(\boldsymbol{W}) \leq \kappa \cdot \mu_{\boldsymbol{X}}(\boldsymbol{W})\}$, we have*

$$\underline{p_{\tilde{\boldsymbol{X}},y}} = \Pr_{\boldsymbol{W} \sim \mu_{\tilde{\boldsymbol{X}}}}[\boldsymbol{W} \in S_\kappa] \quad \text{with } \kappa \in \mathbb{R}_+ \text{ s.t.} \quad \Pr_{\boldsymbol{W} \sim \mu_{\boldsymbol{X}}}[\boldsymbol{W} \in S_\kappa] = p_{\boldsymbol{X},y}$$

*Proof.* See (Neyman and Pearson, 1933) and (Cohen et al., 2019).

There are different views on Lemma 1. In the context of statistical hypothesis testing, Lemma 1 states that the likelihood ratio test is the uniformly most powerful test when testing the simple hypothesis $\mu_{\boldsymbol{X}}(\boldsymbol{W})$ against $\mu_{\tilde{\boldsymbol{X}}}(\boldsymbol{W})$. For such views we refer to Cohen et al. (2019).

Now we prove our main theorem:

**Theorem 1** (Neyman-Pearson lower bound for hierarchical smoothing). *Given fixed $\boldsymbol{X}, \tilde{\boldsymbol{X}} \in \mathcal{X}^{N \times D}$, let $\mu_{\boldsymbol{X}}$ denote a smoothing distribution that operates independently on matrix elements, and $\Psi_{\boldsymbol{X}}$ the induced hierarchical distribution over $\mathcal{Z}^{N \times (D+1)}$. Given label $y \in \mathcal{Y}$ and the probability $p_{\boldsymbol{X},y}$ to classify $\boldsymbol{X}$ as $y$ under $\Psi$. Define $\hat{S}_\kappa \triangleq \{\boldsymbol{W} \in \mathcal{X}^{|\mathcal{C}| \times D} : \mu_{\tilde{\boldsymbol{X}}_\mathcal{C}}(\boldsymbol{W}) \leq \kappa \cdot \mu_{\boldsymbol{X}_\mathcal{C}}(\boldsymbol{W})\}$. We have:*

$$\underline{p_{\tilde{\boldsymbol{X}},y}} = \Pr_{\boldsymbol{W} \sim \mu_{\tilde{\boldsymbol{X}}_\mathcal{C}}}[\boldsymbol{W} \in \hat{S}_\kappa] \cdot (1 - \Delta) \quad \text{with } \kappa \in \mathbb{R}_+ \text{ s.t.} \quad \Pr_{\boldsymbol{W} \sim \mu_{\boldsymbol{X}_\mathcal{C}}}[\boldsymbol{W} \in \hat{S}_\kappa] = \frac{p_{\boldsymbol{X},y} - \Delta}{1 - \Delta}$$

*where $\boldsymbol{X}_\mathcal{C}$ and $\tilde{\boldsymbol{X}}_\mathcal{C}$ denote those rows $\boldsymbol{x}_i$ of $\boldsymbol{X}$ and $\tilde{\boldsymbol{x}}_i$ of $\tilde{\boldsymbol{X}}$ with $i \in \mathcal{C}$, that is $\boldsymbol{x}_i \neq \tilde{\boldsymbol{x}}_i$.*

*Proof.* We are solving the following optimization problem:

$$\underline{p_{\tilde{\boldsymbol{X}},y}} \quad \triangleq \quad \min_{h \in \mathbb{H}} \Pr_{\boldsymbol{Z} \sim \Psi_{\tilde{\boldsymbol{X}}}}[h(\boldsymbol{Z}) = y] \quad s.t. \quad \Pr_{\boldsymbol{Z} \sim \Psi_{\boldsymbol{X}}}[h(\boldsymbol{Z}) = y] = p_{\boldsymbol{X},y}$$

We can use the partitioning of Proposition 1 and apply the law of total probability:

$$\underline{p_{\tilde{\boldsymbol{X}},y}} = \min_{h \in \mathbb{H}} \sum_{i=0}^{3} \Pr_{\boldsymbol{Z} \sim \Psi_{\tilde{\boldsymbol{X}}}}[h(\boldsymbol{Z}) = y \mid \boldsymbol{Z} \in \mathcal{R}_i] \cdot \Pr_{\boldsymbol{Z} \sim \Psi_{\tilde{\boldsymbol{X}}}}[\boldsymbol{Z} \in \mathcal{R}_i]$$

$$s.t. \quad \sum_{i=0}^{3} \Pr_{\boldsymbol{Z} \sim \Psi_{\boldsymbol{X}}}[h(\boldsymbol{Z}) = y \mid \boldsymbol{Z} \in \mathcal{R}_i] \cdot \Pr_{\boldsymbol{Z} \sim \Psi_{\boldsymbol{X}}}[\boldsymbol{Z} \in \mathcal{R}_i] = p_{\boldsymbol{X},y}$$

We apply Proposition 2 and cancel out all zero probabilities:

$$\underline{p_{\tilde{\boldsymbol{X}},y}} = \min_{h \in \mathbb{H}} \Pr_{\boldsymbol{Z} \sim \Psi_{\tilde{\boldsymbol{X}}}}[h(\boldsymbol{Z}) = y \mid \boldsymbol{Z} \in \mathcal{R}_2] \cdot (1 - \Delta) + \Pr_{\boldsymbol{Z} \sim \Psi_{\tilde{\boldsymbol{X}}}}[h(\boldsymbol{Z}) = y \mid \boldsymbol{Z} \in \mathcal{R}_3] \cdot \Delta$$

$$s.t. \quad \Pr_{\boldsymbol{Z} \sim \Psi_{\boldsymbol{X}}}[h(\boldsymbol{Z}) = y \mid \boldsymbol{Z} \in \mathcal{R}_2] \cdot (1 - \Delta) + \Pr_{\boldsymbol{Z} \sim \Psi_{\boldsymbol{X}}}[h(\boldsymbol{Z}) = y \mid \boldsymbol{Z} \in \mathcal{R}_1] \cdot \Delta = p_{\boldsymbol{X},y}$$

This is a minimization problem and to minimize we choose $\Pr_{\boldsymbol{Z} \sim \Psi_{\tilde{\boldsymbol{X}}}}[h(\boldsymbol{Z}) = y \mid \boldsymbol{Z} \in \mathcal{R}_3] = 0$ and $\Pr_{\boldsymbol{Z} \sim \Psi_{\boldsymbol{X}}}[h(\boldsymbol{Z}) = y \mid \boldsymbol{Z} \in \mathcal{R}_1] = 1$. In other words, in the worst-case all matrices in $\mathcal{R}_1$ that we never observe around $\tilde{\boldsymbol{X}}$ are correctly classified and the worst-case classifier first uses the budget $p_{\boldsymbol{X},y}$ on region $\mathcal{R}_1$.

We obtain:

$$\underline{p_{\tilde{\boldsymbol{X}},y}} = \min_{h \in \mathbb{H}} \quad \Pr_{\boldsymbol{Z} \sim \Psi_{\tilde{\boldsymbol{X}}}}[h(\boldsymbol{Z}) = y \mid \boldsymbol{Z} \in \mathcal{R}_2] \cdot (1 - \Delta)$$
$$\text{s.t.} \quad \Pr_{\boldsymbol{Z} \sim \Psi_{\boldsymbol{X}}}[h(\boldsymbol{Z}) = y \mid \boldsymbol{Z} \in \mathcal{R}_2] \cdot (1 - \Delta) = p_{\boldsymbol{X},y} - \Delta$$

Here we can see that for $p = 0$ (i.e. $\Delta = 1$) we have just $\underline{p_{\tilde{\boldsymbol{X}},y}} = 0$. We can assume $p > 0$ in the following.

We can further rearrange the terms:

$$\underline{p_{\tilde{\boldsymbol{X}},y}} = (1 - \Delta) \cdot \min_{h \in \mathbb{H}} \quad \Pr_{\boldsymbol{Z} \sim \Psi_{\tilde{\boldsymbol{X}}}}[h(\boldsymbol{Z}) = y \mid \boldsymbol{Z} \in \mathcal{R}_2]$$
$$\text{s.t.} \quad \Pr_{\boldsymbol{Z} \sim \Psi_{\boldsymbol{X}}}[h(\boldsymbol{Z}) = y \mid \boldsymbol{Z} \in \mathcal{R}_2] = \frac{p_{\boldsymbol{X},y} - \Delta}{1 - \Delta}$$

Now we can apply Lemma 1:

$$\underline{p_{\tilde{\boldsymbol{X}},y}} = \Pr_{\boldsymbol{Z} \sim \Psi_{\tilde{\boldsymbol{X}}}}[\boldsymbol{Z} \in S_\kappa \mid \boldsymbol{Z} \in \mathcal{R}_2] \cdot (1 - \Delta)$$
$$\text{with } \kappa \in \mathbb{R}_+ \text{ s.t.} \tag{2}$$
$$\Pr_{\boldsymbol{Z} \sim \Psi_{\boldsymbol{X}}}[\boldsymbol{Z} \in S_\kappa \mid \boldsymbol{Z} \in \mathcal{R}_2] = \frac{p_{\boldsymbol{X},y} - \Delta}{(1 - \Delta)}$$

for $S_\kappa \triangleq \{\boldsymbol{Z} \in \mathcal{Z}^{N \times (D+1)} : \Psi_{\tilde{\boldsymbol{X}}}(\boldsymbol{Z}) \leq \kappa \cdot \Psi_{\boldsymbol{X}}(\boldsymbol{Z})\}$. Note the difference between $S_\kappa$ and $\hat{S}_\kappa$.

We need to realize that due to independence we have:

$$\{\boldsymbol{Z} \in \mathcal{R}_2 : \Psi_{\tilde{\boldsymbol{X}}}(\boldsymbol{Z}) \leq \kappa \cdot \Psi_{\boldsymbol{X}}(\boldsymbol{Z})\}$$
$$= \{\boldsymbol{Z} \in \mathcal{R}_2 : \Psi_{\tilde{\boldsymbol{X}}}(\boldsymbol{W}, \boldsymbol{\tau}) \leq \kappa \cdot \Psi_{\boldsymbol{X}}(\boldsymbol{W}, \boldsymbol{\tau})\}$$
$$= \{\boldsymbol{Z} \in \mathcal{R}_2 : \mu_{\tilde{\boldsymbol{X}}}(\boldsymbol{W}|\boldsymbol{\tau})\phi(\boldsymbol{\tau}) \leq \kappa \cdot \mu_{\boldsymbol{X}}(\boldsymbol{W}|\boldsymbol{\tau})\phi(\boldsymbol{\tau})\}$$
$$= \left\{\boldsymbol{Z} \in \mathcal{R}_2 : \prod_{i=1}^{N} \mu_{\tilde{\boldsymbol{x}}_i}(\boldsymbol{w}_i|\boldsymbol{\tau}_i) \leq \kappa \cdot \prod_{i=1}^{N} \mu_{\boldsymbol{x}_i}(\boldsymbol{w}_i|\boldsymbol{\tau}_i)\right\} \tag{3}$$
$$\overset{(*)}{=} \left\{\boldsymbol{Z} \in \mathcal{R}_2 : \prod_{i \in \mathcal{C}} \mu_{\tilde{\boldsymbol{x}}_i}(\boldsymbol{w}_i) \leq \kappa \cdot \prod_{i \in \mathcal{C}} \mu_{\boldsymbol{x}_i}(\boldsymbol{w}_i)\right\}$$
$$= \{\boldsymbol{Z} \in \mathcal{R}_2 : \mu_{\tilde{\boldsymbol{X}}_\mathcal{C}}(\boldsymbol{W}_\mathcal{C}) \leq \kappa \cdot \mu_{\boldsymbol{X}_\mathcal{C}}(\boldsymbol{W}_\mathcal{C})\}$$

where (*) is due to $\boldsymbol{x}_i = \tilde{\boldsymbol{x}}_i$ for all $i \notin \mathcal{C}$ and $\boldsymbol{\tau}_i = 1$ for $i \in \mathcal{C}$.

In other words this means that if $\boldsymbol{Z} \in S_\kappa \cap \mathcal{R}_2$ then all $\boldsymbol{Z}' \in \mathcal{R}_2$ that differ only in rows $i \notin \mathcal{C}$ from $\boldsymbol{Z}$ are also in $S_\kappa \cap \mathcal{R}_2$ and integrate out (the "likelihood ratio" is only determined by rows $i \in \mathcal{C}$).

It follows that for fixed $\kappa \in \mathbb{R}_+$ we have

$$\Pr_{\boldsymbol{Z} \sim \Psi_{\tilde{\boldsymbol{X}}}}[\boldsymbol{Z} \in S_\kappa \mid \boldsymbol{Z} \in \mathcal{R}_2] = \Pr_{\boldsymbol{W} \sim \mu_{\tilde{\boldsymbol{X}}_\mathcal{C}}}[\boldsymbol{W} \in \hat{S}_\kappa]$$

and

$$\Pr_{\boldsymbol{Z} \sim \Psi_{\boldsymbol{X}}}[\boldsymbol{Z} \in S_\kappa \mid \boldsymbol{Z} \in \mathcal{R}_2] = \Pr_{\boldsymbol{W} \sim \mu_{\boldsymbol{X}_\mathcal{C}}}[\boldsymbol{W} \in \hat{S}_\kappa]$$

for $\hat{S}_\kappa \triangleq \{\boldsymbol{W} \in \mathcal{X}^{|\mathcal{C}| \times D} : \mu_{\tilde{\boldsymbol{X}}_\mathcal{C}}(\boldsymbol{W}) \leq \kappa \cdot \mu_{\boldsymbol{X}_\mathcal{C}}(\boldsymbol{W})\}$. All together we obtain:

$$\underline{p_{\tilde{\boldsymbol{X}},y}} = \Pr_{\boldsymbol{W} \sim \mu_{\tilde{\boldsymbol{X}}_\mathcal{C}}}[\boldsymbol{W} \in \hat{S}_\kappa] \cdot (1 - \Delta) \quad \text{with } \kappa \in \mathbb{R}_+ \text{ s.t.} \quad \Pr_{\boldsymbol{W} \sim \mu_{\boldsymbol{X}_\mathcal{C}}}[\boldsymbol{W} \in \hat{S}_\kappa] = \frac{p_{\boldsymbol{X},y} - \Delta}{1 - \Delta}$$

$$\square$$

# D Multi-class certificates for hierarchical randomized smoothing

In the main section of our paper we only derive so-called binary-class certificates: If $p_{\tilde{\boldsymbol{X}},y^*} > 0.5$, then the smoothed classifier $g$ is certifiably robust. We can derive a tighter certificate by guaranteeing $\underline{p_{\tilde{\boldsymbol{X}},y^*}} > \max_{y \neq y^*} \overline{p_{\tilde{\boldsymbol{X}},y}}$. To this end, we derive an upper bound $p_{\tilde{\boldsymbol{X}},y} \leq \overline{p_{\tilde{\boldsymbol{X}},y}}$ in the following:

$$\overline{p_{\tilde{\boldsymbol{X}},y}} \triangleq \max_{h \in \mathbb{H}} \Pr_{\boldsymbol{W} \sim \mu_{\tilde{\boldsymbol{X}}}} [h(\boldsymbol{W}) = y] \quad s.t. \quad \Pr_{\boldsymbol{W} \sim \mu_{\boldsymbol{X}}} [h(\boldsymbol{W}) = y] = p_{\boldsymbol{X},y} \tag{4}$$

Cohen et al. (2019) show that the maximum of Equation 4 can be obtained by using following lemma:

**Lemma 3** (Neyman-Pearson upper bound). *Given* $\boldsymbol{X}, \tilde{\boldsymbol{X}} \in \mathcal{X}^{N \times D}$, *distributions* $\mu_{\boldsymbol{X}}, \mu_{\tilde{\boldsymbol{X}}}$, *class label* $y \in \mathcal{Y}$, *probability* $p_{\boldsymbol{X},y}$ *and the set* $S_\kappa \triangleq \left\{ \boldsymbol{W} \in \mathcal{X}^{N \times D} : \mu_{\tilde{\boldsymbol{X}}}(\boldsymbol{W}) \geq \kappa \cdot \mu_{\boldsymbol{X}}(\boldsymbol{W}) \right\}$ *we have*

$$\overline{p_{\tilde{\boldsymbol{X}},y}} = \Pr_{\boldsymbol{W} \sim \mu_{\tilde{\boldsymbol{X}}}} [\boldsymbol{W} \in S_\kappa] \quad \text{with } \kappa \in \mathbb{R}_+ \text{ s.t.} \quad \Pr_{\boldsymbol{W} \sim \mu_{\boldsymbol{X}}} [\boldsymbol{W} \in S_\kappa] = p_{\boldsymbol{X},y}$$

*Proof.* See (Neyman and Pearson, 1933; Cohen et al., 2019). $\square$

We can now use Lemma 3 to derive an upper bound under hierarchical smoothing:

**Theorem 2** (Neyman-Pearson upper bound for hierarchical smoothing). *Given fixed* $\boldsymbol{X}, \tilde{\boldsymbol{X}} \in \mathcal{X}^{N \times D}$, *let* $\mu_{\boldsymbol{X}}$ *denote a smoothing distribution that operates independently on matrix elements, and* $\Psi_{\boldsymbol{X}}$ *the induced hierarchical distribution over* $\mathcal{Z}^{N \times (D+1)}$. *Given label* $y \in \mathcal{Y}$ *and the probability* $p_{\boldsymbol{X},y}$ *to classify* $\boldsymbol{X}$ *as* $y$ *under* $\Psi$. *Define* $\hat{S}_\kappa \triangleq \left\{ \boldsymbol{W} \in \mathcal{X}^{r \times D} : \mu_{\tilde{\boldsymbol{X}}_{\mathcal{C}}}(\boldsymbol{W}) \geq \kappa \cdot \mu_{\boldsymbol{X}_{\mathcal{C}}}(\boldsymbol{W}) \right\}$. *We have:*

$$\overline{p_{\tilde{\boldsymbol{X}},y}} = \Pr_{\boldsymbol{W} \sim \mu_{\tilde{\boldsymbol{X}}_{\mathcal{C}}}} [\boldsymbol{W} \in \hat{S}_\kappa] \cdot (1 - \Delta) + \Delta \quad \text{with } \kappa \in \mathbb{R}_+ \text{ s.t.} \quad \Pr_{\boldsymbol{W} \sim \mu_{\boldsymbol{X}_{\mathcal{C}}}} [\boldsymbol{W} \in \hat{S}_\kappa] = \frac{p_{\boldsymbol{X},y}}{1 - \Delta}$$

*where* $\boldsymbol{X}_{\mathcal{C}}$ *and* $\tilde{\boldsymbol{X}}_{\mathcal{C}}$ *denote those rows* $\boldsymbol{x}_i$ *of* $\boldsymbol{X}$ *and* $\tilde{\boldsymbol{x}}_i$ *of* $\tilde{\boldsymbol{X}}$ *with* $i \in \mathcal{C}$, *that is* $\boldsymbol{x}_i \neq \tilde{\boldsymbol{x}}_i$.

*Proof.* We are solving the following maximization problem:

$$\overline{p_{\tilde{\boldsymbol{X}},y}} \triangleq \max_{h \in \mathbb{H}} \Pr_{\boldsymbol{Z} \sim \Psi_{\tilde{\boldsymbol{X}}}} [h(\boldsymbol{Z}) = y] \quad s.t. \quad \Pr_{\boldsymbol{Z} \sim \Psi_{\boldsymbol{X}}} [h(\boldsymbol{Z}) = y] = p_{\boldsymbol{X},y}$$

We can use the partitioning of Proposition 1 and apply the law of total probability:

$$\overline{p_{\tilde{\boldsymbol{X}},y}} = \max_{h \in \mathbb{H}} \sum_{i=0}^{3} \Pr_{\boldsymbol{Z} \sim \Psi_{\tilde{\boldsymbol{X}}}} [h(\boldsymbol{Z}) = y \mid \boldsymbol{Z} \in \mathcal{R}_i] \cdot \Pr_{\boldsymbol{Z} \sim \Psi_{\tilde{\boldsymbol{X}}}} [\boldsymbol{Z} \in \mathcal{R}_i]$$

$$\text{s.t.} \quad \sum_{i=0}^{3} \Pr_{\boldsymbol{Z} \sim \Psi_{\boldsymbol{X}}} [h(\boldsymbol{Z}) = y \mid \boldsymbol{Z} \in \mathcal{R}_i] \cdot \Pr_{\boldsymbol{Z} \sim \Psi_{\boldsymbol{X}}} [\boldsymbol{Z} \in \mathcal{R}_i] = p_{\boldsymbol{X},y}$$

We apply Proposition 2 and cancel out all zero probabilities:

$$\overline{p_{\tilde{\boldsymbol{X}},y}} = \max_{h \in \mathbb{H}} \Pr_{\boldsymbol{Z} \sim \Psi_{\tilde{\boldsymbol{X}}}} [h(\boldsymbol{Z}) = y \mid \boldsymbol{Z} \in \mathcal{R}_2] \cdot (1 - \Delta) + \Pr_{\boldsymbol{Z} \sim \Psi_{\tilde{\boldsymbol{X}}}} [h(\boldsymbol{Z}) = y \mid \boldsymbol{Z} \in \mathcal{R}_3] \cdot \Delta$$

$$\text{s.t.} \quad \Pr_{\boldsymbol{Z} \sim \Psi_{\boldsymbol{X}}} [h(\boldsymbol{Z}) = y \mid \boldsymbol{Z} \in \mathcal{R}_2] \cdot (1 - \Delta) + \Pr_{\boldsymbol{Z} \sim \Psi_{\boldsymbol{X}}} [h(\boldsymbol{Z}) = y \mid \boldsymbol{Z} \in \mathcal{R}_1] \cdot \Delta = p_{\boldsymbol{X},y}$$

This is a maximization problem and to maximize we choose $\Pr_{\boldsymbol{Z} \sim \Psi_{\tilde{\boldsymbol{X}}}}[h(\boldsymbol{Z}) = y \mid \boldsymbol{Z} \in \mathcal{R}_3] = 1$ and $\Pr_{\boldsymbol{Z} \sim \Psi_{\boldsymbol{X}}}[h(\boldsymbol{Z}) = y \mid \boldsymbol{Z} \in \mathcal{R}_1] = 0$. In other words, all matrices in $\mathcal{R}_3$ that we never observe around $\boldsymbol{X}$ are correctly classified and the worst-case classifier does not use any budget $p_{\boldsymbol{X},y}$ on this region $\mathcal{R}_3$.

We obtain:

$$\overline{p_{\tilde{\boldsymbol{X}},y}} = \max_{h \in \mathbb{H}} \Pr_{\boldsymbol{Z} \sim \Psi_{\tilde{\boldsymbol{X}}}} [h(\boldsymbol{Z}) = y \mid \boldsymbol{Z} \in \mathcal{R}_2] \cdot (1 - \Delta) + \Delta$$

$$\text{s.t.} \quad \Pr_{\boldsymbol{Z} \sim \Psi_{\boldsymbol{X}}} [h(\boldsymbol{Z}) = y \mid \boldsymbol{Z} \in \mathcal{R}_2] \cdot (1 - \Delta) = p_{\boldsymbol{X},y}$$

The remaining statement follows analogous to the proof in Appendix C. $\square$

In this section we derived multi-class robustness certificates for randomized smoothing and showed the correctness of the following certification procedure:

---

**Algorithm 2:** Multi-class Robustness Certificates for Hierarchical Randomized Smoothing

---

**Input:** $f$, $\boldsymbol{X}$, $p$, Lower-level smoothing distribution $\mu$, Radius $(r, \epsilon)$, $n_0$, $n_1$, $1 - \alpha$

counts0 $\leftarrow$ SampleUnderNoise($f$, $\boldsymbol{X}$, $p$, $\mu$, $n_0$) ;

counts1 $\leftarrow$ SampleUnderNoise($f$, $\boldsymbol{X}$, $p$, $\mu$, $n_1$) ;

$y_A, y_B \leftarrow$ top two indices in counts0 ;

$\underline{p_A}, \overline{p_B} \leftarrow$ ConfidenceBounds(counts1[$y_A$],counts1[$y_B$], $n_1$, $1 - \alpha$) ;

$\Delta \leftarrow 1 - p^r$ ;

$\overline{\tilde{p}_B} \leftarrow$ LowerLevelWorstCaseUpperBound($\mu$, $\epsilon$, $\frac{\overline{p_B}}{1-\Delta}$) ;

$\underline{\tilde{p}_A} \leftarrow$ LowerLevelWorstCaseLowerBound($\mu$, $\epsilon$, $\frac{\underline{p_A}-\Delta}{1-\Delta}$) ;

**if** $\underline{\tilde{p}_A} \cdot (1 - \Delta) > \overline{\tilde{p}_B} \cdot (1 - \Delta) + \Delta$ **then**

  | **Return:** $y_A$ certified

**end**

**Return:** ABSTAIN

---

Here, LowerLevelWorstCaseUpperBound($\mu$,$\epsilon$, $\overline{p_B}$) computes the existing certificate, specifically the largest $\overline{p_{\tilde{\boldsymbol{X}}, y_B}}$ over the entire threat model, e.g. $\Phi\left(\Phi^{-1}(\overline{p_B}) + \frac{\epsilon}{\sigma}\right)$ for Gaussian smoothing. The remaining certification procedure is analogous to the certification in (Cohen et al., 2019).

# E Hierarchical randomized smoothing using Gaussian isotropic smoothing (Proofs of Section 6)

Having derived our general certification framework for hierarchical randomized smoothing, we can instantiate it with specific smoothing distributions to certify specific threat models. In the following we show the robustness guarantees for initializing hierarchical randomized smoothing with Gaussian isotropic smoothing, as already discussed in Section 6. We will also derive the corresponding multi-class robustness certificates.

**Corollary 3.** *Given continuous $\mathcal{X} \triangleq \mathbb{R}$, consider the threat model $\mathcal{B}_{2,\epsilon}^r(\boldsymbol{X})$ for fixed $\boldsymbol{X} \in \mathcal{X}^{N \times D}$. Initialize hierarchical smoothing with the isotropic Gaussian smoothing distribution defined as $\mu_{\boldsymbol{X}}(\boldsymbol{W}) \triangleq \prod_{i=1}^{N} \mathcal{N}(\boldsymbol{w}_i | \boldsymbol{x}_i, \sigma^2 \boldsymbol{I})$. Let $y \in \mathcal{Y}$ denote a label and $p_{\boldsymbol{X},y}$ the probability to classify $\boldsymbol{X}$ as $y$ under $\Psi$. Define $\Delta \triangleq 1 - p^r$. Then we have:*

$$\min_{\tilde{\boldsymbol{X}} \in \mathcal{B}_{2,\epsilon}^r(\boldsymbol{X})} \underline{p_{\tilde{\boldsymbol{X}},y}} = \Phi\left(\Phi^{-1}\left(\frac{p_{\boldsymbol{X},y} - \Delta}{1 - \Delta}\right) - \frac{\epsilon}{\sigma}\right)(1 - \Delta)$$

*Proof.* As outlined in Subsection 3.3, Cohen et al. (2019) show that the optimal value of Equation 1 under Gaussian smoothing is $\Phi\left(\Phi^{-1}(p_{\boldsymbol{x},y}) - \frac{||\boldsymbol{\delta}||_2}{\sigma}\right)$, where $\Phi$ denotes the CDF of the normal distribution and $\boldsymbol{\delta} = \boldsymbol{x} - \tilde{\boldsymbol{x}}$. Plugging this optimal value into Theorem 1 yields for any $\tilde{\boldsymbol{X}} \in \mathcal{B}_{2,\epsilon}^r(\boldsymbol{X})$:

$$\underline{p_{\tilde{\boldsymbol{X}},y}} = \Phi\left(\Phi^{-1}\left(\frac{p_{\boldsymbol{X},y} - \Delta}{1 - \Delta}\right) - \frac{||\boldsymbol{\delta}||_2}{\sigma}\right)(1 - \Delta)$$

where $\Delta = 1 - p^{|\mathcal{C}|}$ and $\boldsymbol{\delta} = vec(\boldsymbol{X} - \tilde{\boldsymbol{X}})$. The remaining statements about the minimum over the entire threat model follows from the fact that $\underline{p_{\tilde{\boldsymbol{X}},y}}$ is monotonously decreasing in $||\boldsymbol{\delta}||_2$, and monotonously decreasing in $r$ (see also Proposition 3 in Section 5). Note that for fixed $\tilde{\boldsymbol{X}}$ we write $\Delta = 1 - p^{|\mathcal{C}|}$ but for the entire threat model we have to use the largest $\Delta = 1 - p^r$. $\square$

**Corollary 1.** *Given continuous $\mathcal{X} \triangleq \mathbb{R}$, consider the threat model $\mathcal{B}_{2,\epsilon}^r(\boldsymbol{X})$ for fixed $\boldsymbol{X} \in \mathcal{X}^{N \times D}$. Initialize the hierarchical smoothing distribution $\Psi$ using the isotropic Gaussian distribution $\mu_{\boldsymbol{X}}(\boldsymbol{W}) \triangleq \prod_{i=1}^{N} \mathcal{N}(\boldsymbol{w}_i | \boldsymbol{x}_i, \sigma^2 \boldsymbol{I})$ that applies noise independently on each matrix element. Let $y^* \in \mathcal{Y}$ denote the majority class and $p_{\boldsymbol{X},y^*}$ the probability to classify $\boldsymbol{X}$ as $y^*$ under $\Psi$. Then the smoothed classifier $g$ is certifiably robust $g(\boldsymbol{X}) = g(\tilde{\boldsymbol{X}})$ for any $\tilde{\boldsymbol{X}} \in \mathcal{B}_{2,\epsilon}^r(\boldsymbol{X})$ if*

$$\epsilon < \sigma\left(\Phi^{-1}\left(\frac{p_{\boldsymbol{X},y^*} - \Delta}{1 - \Delta}\right) - \Phi^{-1}\left(\frac{1}{2(1 - \Delta)}\right)\right)$$

*where $\Phi^{-1}$ denotes the inverse CDF of the normal distribution and $\Delta \triangleq 1 - p^r$.*

*Proof.* For the binary-class certificate, the smoothed classifier $g$ is certifiably robust for fixed input $\boldsymbol{X}$ if $\min_{\tilde{\boldsymbol{X}} \in \mathcal{B}_{2,\epsilon}^r(\boldsymbol{X})} \underline{p_{\tilde{\boldsymbol{X}},y^*}} > 0.5$. By using Corollary 3 we have

$$\min_{\tilde{\boldsymbol{X}} \in \mathcal{B}_{2,\epsilon}^r(\boldsymbol{X})} \underline{p_{\tilde{\boldsymbol{X}},y^*}} > \frac{1}{2}$$

$$\Leftrightarrow \Phi\left(\Phi^{-1}\left(\frac{p_{\boldsymbol{X},y^*} - \Delta}{1 - \Delta}\right) - \frac{\epsilon}{\sigma}\right)(1 - \Delta) > \frac{1}{2}$$

$$\Leftrightarrow \Phi^{-1}\left(\frac{p_{\boldsymbol{X},y^*} - \Delta}{1 - \Delta}\right) - \frac{\epsilon}{\sigma} > \Phi^{-1}\left(\frac{1}{2(1 - \Delta)}\right)$$

$$\Leftrightarrow \epsilon < \sigma\left(\Phi^{-1}\left(\frac{p_{\boldsymbol{X},y^*} - \Delta}{1 - \Delta}\right) - \Phi^{-1}\left(\frac{1}{2(1 - \Delta)}\right)\right)$$

$\square$

In the following we additionally derive the corresponding multi-class robustness certificates:

**Corollary 4.** *Given continuous $\mathcal{X} \triangleq \mathbb{R}$, consider the threat model $\mathcal{B}_{2,\epsilon}^r(\boldsymbol{X})$ for fixed $\boldsymbol{X} \in \mathcal{X}^{N \times D}$. Initialize hierarchical smoothing with the isotropic Gaussian smoothing distribution defined as $\mu_{\boldsymbol{X}}(\boldsymbol{W}) \triangleq \prod_{i=1}^{N} \mathcal{N}(\boldsymbol{w}_i | \boldsymbol{x}_i, \sigma^2 \boldsymbol{I})$. Let $y \in \mathcal{Y}$ denote a label and $p_{\boldsymbol{X},y}$ the probability to classify $\boldsymbol{X}$ as $y$ under $\Psi$. Define $\Delta \triangleq 1 - p^r$. Then we have:*

$$\max_{\tilde{\boldsymbol{X}} \in \mathcal{B}_{2,\epsilon}^r(\boldsymbol{X})} \overline{p_{\tilde{\boldsymbol{X}},y}} = \Phi\left(\Phi^{-1}\left(\frac{p_{\boldsymbol{X},y}}{1-\Delta}\right) + \frac{\epsilon}{\sigma}\right)(1-\Delta) + \Delta$$

*Proof.* Cohen et al. (2019) show that the optimal value of Equation 4 under Gaussian smoothing is $\Phi\left(\Phi^{-1}(p_{\boldsymbol{x},y}) + \frac{\epsilon}{\sigma}\right)$, where $\Phi$ denotes the CDF of the normal distribution. The statement follows analogously to Corollary 3 by directly plugging this optimal value into Theorem 2. $\qquad\square$

**Corollary 5** (Multi-class certificates). *Given continuous $\mathcal{X} \triangleq \mathbb{R}$, consider the threat model $\mathcal{B}_{2,\epsilon}^r(\boldsymbol{X})$ for fixed $\boldsymbol{X} \in \mathcal{X}^{N \times D}$. Initialize the hierarchical smoothing distribution $\Psi$ using the isotropic Gaussian distribution $\mu_{\boldsymbol{X}}(\boldsymbol{W}) \triangleq \prod_{i=1}^{N} \mathcal{N}(\boldsymbol{w}_i | \boldsymbol{x}_i, \sigma^2 \boldsymbol{I})$ that applies noise independently on each matrix element. Given majority class $y^* \in \mathcal{Y}$ and the probability $p_{\boldsymbol{X},y^*}$ to classify $\boldsymbol{X}$ as $y^*$ under $\Psi$. Then the smoothed classifier $g$ is certifiably robust $g(\boldsymbol{X}) = g(\tilde{\boldsymbol{X}})$ for any $\tilde{\boldsymbol{X}} \in \mathcal{B}_{2,\epsilon}^r(\boldsymbol{X})$ if*

$$\Phi\left(\Phi^{-1}\left(\frac{p_{\boldsymbol{X},y^*} - \Delta}{1-\Delta}\right) - \frac{\epsilon}{\sigma}\right) > \max_{y \neq y^*} \Phi\left(\Phi^{-1}\left(\frac{p_{\boldsymbol{X},y}}{1-\Delta}\right) + \frac{\epsilon}{\sigma}\right) + \frac{\Delta}{1-\Delta}$$

*where $\Phi^{-1}$ denotes the inverse CDF of the normal distribution and $\Delta \triangleq 1 - p^r$.*

*Proof.* For the tighter multi-class certificate we have to guarantee $\underline{p_{\tilde{\boldsymbol{X}},y^*}} > \max_{y \neq y^*} \overline{p_{\tilde{\boldsymbol{X}},y}}$. The statement follows directly from Corollary 3 and Corollary 4. $\qquad\square$

# F  Hierarchical smoothing for discrete lower-level smoothing distributions

**Threat model details for discrete hierarchical smoothing experiments.** In our discrete experiments in Section 7 we model adversaries that perturb multiple nodes in the graph by deleting $r_d$ ones (flip $1 \to 0$) and adding $r_a$ ones (flip $0 \to 1$) in the feature matrix $\boldsymbol{X}$, that is we consider the ball $\mathcal{B}^r_{r_a, r_d}(\boldsymbol{X})$. Here we define this threat model more formally:

$$\mathcal{B}^r_{r_a, r_d}(\boldsymbol{X}) \triangleq \{\tilde{\mathbf{X}} \in \mathcal{X}^{N \times D} \mid \sum_{i=1}^{N} \mathbb{1}[\mathbf{x}_i \neq \tilde{\mathbf{x}}_i] \leq r$$
$$\sum_i \sum_j \mathbb{1}(\tilde{\boldsymbol{X}}_{ij} = \boldsymbol{X}_{ij} + 1) \leq r_a \qquad (5)$$
$$\sum_i \sum_j \mathbb{1}(\tilde{\boldsymbol{X}}_{ij} = \boldsymbol{X}_{ij} - 1) \leq r_d\}$$

where $\mathbb{1}$ is an indicator function, $r$ the number of controlled nodes, $r_a$ the number of adversarial insertions ($0 \to 1$), and $r_d$ the number of adversarial deletions ($1 \to 0$).

We expand the result of Theorem 1 to certificates for discrete lower-level smoothing distributions $\mu$:

**Theorem 3** (Hierarchical lower bound for discrete lower-level smoothing distributions)**.** *Given discrete $\mathcal{X}$ and a smoothing distribution $\mu_{\boldsymbol{x}}$ for discrete data, assume that the corresponding discrete certificate partitions the input space $\mathcal{X}^{rD}$ into disjoint regions $\mathcal{H}_1, \ldots, \mathcal{H}_I$ of constant likelihood ratio: $\mu_{\boldsymbol{x}}(\boldsymbol{w})/\mu_{\tilde{\boldsymbol{x}}}(\boldsymbol{w}) = c_i$ for all $\boldsymbol{w} \in \mathcal{H}_i$. Then we can compute a lower bound $\underline{p_{\tilde{\boldsymbol{X}}, y}} \leq p_{\tilde{\boldsymbol{X}}, y}$ by solving the following Linear Program (LP):*

$$\underline{p_{\tilde{\boldsymbol{X}}, y}} = \min_{\boldsymbol{h}} \quad \boldsymbol{h}^T \tilde{\boldsymbol{\nu}} \cdot (1 - \Delta) \quad s.t. \quad \boldsymbol{h}^T \boldsymbol{\nu} = \frac{p_{\boldsymbol{X}, y} - \Delta}{1 - \Delta}, \quad 0 \leq \boldsymbol{h}_i \leq 1$$

*where $\tilde{\boldsymbol{\nu}}_q \triangleq \Pr_{\boldsymbol{W} \sim \mu_{\tilde{\boldsymbol{X}}_C}}[vec(\boldsymbol{W}) \in \mathcal{H}_q]$ and $\boldsymbol{\nu}_q \triangleq \Pr_{\boldsymbol{W} \sim \mu_{\boldsymbol{X}_C}}[vec(\boldsymbol{W}) \in \mathcal{H}_q]$ for all $q \in \{1, \ldots, I\}$.*

*Proof.* This follows from applying the Neyman-Pearson-Lemma for discrete random variables twice (see Lemma 2). While $\mathcal{R}_1, \mathcal{R}_2, \mathcal{R}_3$ represent "super regions" for the upper-level smoothing distribution $\phi$, $\mathcal{H}_1, \ldots, \mathcal{H}_I$ further subdivide $\mathcal{R}_2$ into smaller regions of constant likelihood ratio, which is required for the lower-level discrete smoothing distribution $\mu$. $\qquad \square$

Notably, Theorem 3 implies that we can compute the LP of the original discrete certificate for the adjusted probability of $(p_{\boldsymbol{X}, y} - \Delta)/(1 - \Delta)$, and then multiply the resulting optimal value by the constant $1 - \Delta$. In particular, the feasibility of the Linear Program and of computing $\boldsymbol{v}, \tilde{\boldsymbol{v}}$ can be delegated to the certificate for the underlying $\mu$.

Analogously for the upper bound for discrete lower-level $\mu$ we need:

**Lemma 4** (Discrete Neyman-Pearson upper bound)**.** *Partition the input space $\mathcal{X}^D$ into disjoint regions $\mathcal{R}_1, \ldots, \mathcal{R}_I$ of constant likelihood ratio $\mu_{\boldsymbol{x}}(\boldsymbol{w})/\mu_{\tilde{\boldsymbol{x}}}(\boldsymbol{w}) = c_i \in \mathbb{R} \cup \{\infty\}$ for all $\boldsymbol{w} \in \mathcal{R}_i$. Define $\tilde{\boldsymbol{r}}_i \triangleq \Pr_{\boldsymbol{w} \sim \mu_{\tilde{\boldsymbol{x}}}}(\boldsymbol{w} \in \mathcal{R}_i)$ and $\boldsymbol{r}_i \triangleq \Pr_{\boldsymbol{w} \sim \mu_{\boldsymbol{x}}}(\boldsymbol{w} \in \mathcal{R}_i)$. Then Equation 4 is equivalent to the linear program $\overline{p_{\tilde{\boldsymbol{x}}, y}} = \max_{\boldsymbol{h} \in [0,1]^I} \boldsymbol{h}^T \tilde{\boldsymbol{r}}$ s.t. $\boldsymbol{h}^T \boldsymbol{r} = p_{\boldsymbol{x}, y}$, where $\boldsymbol{h}$ represents the worst-case classifier.*

Then we can show:

**Theorem 4** (Hierarchical upper bound for discrete lower-level smoothing distributions)**.** *Given discrete $\mathcal{X}$ and a smoothing distribution $\mu_{\boldsymbol{x}}$ for discrete data, assume that the corresponding discrete certificate partitions the input space $\mathcal{X}^{rD}$ into disjoint regions $\mathcal{H}_1, \ldots, \mathcal{H}_I$ of constant likelihood ratio: $\mu_{\boldsymbol{x}}(\boldsymbol{w})/\mu_{\tilde{\boldsymbol{x}}}(\boldsymbol{w}) = c_i$ for all $\boldsymbol{w} \in \mathcal{H}_i$. Then we can compute an upper bound $\overline{p_{\tilde{\boldsymbol{X}}, y}} \geq p_{\tilde{\boldsymbol{X}}, y}$ by solving the following Linear Program (LP):*

$$\overline{p_{\tilde{\boldsymbol{X}}, y}} = \max_{\boldsymbol{h}} \quad \boldsymbol{h}^T \tilde{\boldsymbol{\nu}} \cdot (1 - \Delta) + \Delta \quad s.t. \quad \boldsymbol{h}^T \boldsymbol{\nu} = \frac{p_{\boldsymbol{X}, y}}{1 - \Delta}, \quad 0 \leq \boldsymbol{h}_i \leq 1$$

*where $\tilde{\boldsymbol{\nu}}_q \triangleq \Pr_{\boldsymbol{W} \sim \mu_{\tilde{\boldsymbol{X}}_C}}[vec(\boldsymbol{W}) \in \mathcal{H}_q]$ and $\boldsymbol{\nu}_q \triangleq \Pr_{\boldsymbol{W} \sim \mu_{\boldsymbol{X}_C}}[vec(\boldsymbol{W}) \in \mathcal{H}_q]$ for all $q \in \{1, \ldots, I\}$.*

*Proof.* This follows analogously to Theorem 3, specifically by applying the Neyman-Pearson-Lemma upper bound for discrete random variables (Lemma 4) twice (first using the "super regions" and then the "lower-level" regions). $\qquad \square$

# G   Hierarchical randomized smoothing using ablation smoothing

There are different types randomized ablation certificates: In image classification, Levine and Feizi (2020b) randomly ablate pixels by masking them. In point cloud classification, Liu et al. (2021) ablate points by deleting them to certify robustness against adversarial point insertion, deletion and modification. In sequence classification, Huang et al. (2023) randomly delete sequence elements to certify robustness against edit distance threat models. In node/graph classification, Bojchevski et al. (2020) experiment with ablation smoothing for individual attributes but show that their additive method is superior. Scholten et al. (2022) directly mask out entire nodes, yielding strong certificates against adversaries that arbitrarily perturb node attributes.

**Ablation smoothing differences.** The ablation baseline (Levine and Feizi, 2020b) draws $k$ pixels to ablate from a uniform distribution $\mathcal{U}(k, N)$ over all possible subsets with $k$ out of $N$ pixels. Levine and Feizi (2020b) derive $\Delta^L = 1 - \frac{\binom{N-r}{k}}{\binom{N}{k}}$. This is in contrast to our approach, which selects pixels to add noise to with a probability of $p$, for which we have $\Delta^S = 1 - p^r$. In other words, we choose the pixel selection probability of $p = 1 - \frac{k}{N}$ (i.e. we select $k$ pixels in expectation). This way, increasing $k$ corresponds to decreasing $p$ and vice versa. Interestingly, this type of ablation smoothing is tighter for small $N$ since $\Delta^S < \Delta^L$ for $k > 0$ (see Scholten et al. (2022) (Appendix B, Proposition 5) for a detailed discussion; later also verified by Huang et al. (2023)).

**Ablation and additive noise hybrid.** Notably, our method generalizes ablation and additive noise smoothing into a common framework: The additional indicator indicates which entity will be ablated, but instead of ablating entities we only add noise to them. If we ablate the selected rows instead of adding noise, we can restore certificates for ablation smoothing as a special case of our framework:

**Corollary 2.** *Initialize the hierarchical smoothing distribution $\Psi$ with the ablation smoothing distribution $\mu_{\boldsymbol{x}}(\boldsymbol{t}) = 1$ for ablation token $\boldsymbol{t} \notin \mathcal{X}^D$ and $\mu_{\boldsymbol{x}}(\boldsymbol{w}) = 0$ for $\boldsymbol{w} \in \mathcal{X}^D$ otherwise (i.e. we always ablate selected rows). Define $\Delta \triangleq 1 - p^r$. Then the smoothed classifier $g$ is certifiably robust $g(\boldsymbol{X}) = g(\tilde{\boldsymbol{X}})$ for any $\tilde{\boldsymbol{X}} \in \mathcal{B}_{2,\epsilon}^r(\boldsymbol{X})$ if $p_{\boldsymbol{X},y} - \Delta > 0.5$.*

*Proof.* We can first apply Theorem 3 and obtain:

$$\underline{p_{\tilde{\boldsymbol{X}},y}} = min_{\boldsymbol{h}} \quad \boldsymbol{h}^T \tilde{\boldsymbol{\nu}} \cdot (1 - \Delta) \quad \text{s.t.} \quad \boldsymbol{h}^T \boldsymbol{\nu} = \frac{p_{\boldsymbol{X},y} - \Delta}{1 - \Delta}, \quad 0 \le \boldsymbol{h}_i \le 1$$

for $\tilde{\boldsymbol{v}} = 1$ and $\boldsymbol{v} = 1$ (since we always ablate selected rows there is only a single lower-level region $\mathcal{H}$). Solving this LP yields

$$\underline{p_{\tilde{\boldsymbol{X}},y}} = \frac{p_{\boldsymbol{X},y} - \Delta}{1 - \Delta}(1 - \Delta) = p_{\boldsymbol{X},y} - \Delta$$

$\square$

Note one can analogously compute the upper bound $\overline{p_{\tilde{\boldsymbol{X}},y}}$ by applying the corresponding Theorem 4:

$$\overline{p_{\tilde{\boldsymbol{X}},y}} = \frac{p_{\boldsymbol{X},y}}{1 - \Delta}(1 - \Delta) + \Delta = p_{\boldsymbol{X},y} + \Delta$$

Altogether in this special case we obtain for the multi-class certificate $\underline{p_{\tilde{\boldsymbol{X}},y^*}} > \max_{y \neq y^*} \overline{p_{\tilde{\boldsymbol{X}},y}}$:

$$p_{\boldsymbol{X},y^*} - \Delta > p_{\boldsymbol{X},\tilde{y}} + \Delta$$

where $y^*$ is the most likely class and $\tilde{y}$ the follow-up class. That is in this special case with ablation smoothing we again obtain the multi-class guarantees derived by Levine and Feizi (2020b).

**Related work.** In the context of randomized ablation (a special case of our framework), Jia et al. (2021) also define three regions and build an ablation certificate using the Neyman-Pearson Lemma. Their regions are, however, different to ours, and their certificates are designed for poisoning threat models (whereas we consider evasion settings). Moreover, Liu et al. (2021) define three regions similar to ours, but their certificate is only for point cloud classifiers and suffers from the same limitations of ablation certificates that we overcome: Our key idea is to further process the region $\mathcal{R}_2$ and to utilize the budget $p_{\boldsymbol{X},y} - \Delta$ by plugging it into the certificate for the lower-level smoothing distribution, which yields stronger robustness-accuracy trade-offs under our threat model.

**Discussion.** In general, the difference between ablation certificates and additive noise certificates is that ablation smoothing does not evaluate the base classifier on the entire input space. Specifically, the supported sample spaces around $X$ and $\tilde{X}$ are overlapping but not the identical. In this context, additive noise certificates are technically tighter for threat models where adversarial perturbations are bounded. Since our certificates generalize ablation smoothing and additive noise smoothing we inherit advantages and disadvantages of both methods: While the certificate for the lower-level smoothing distribution may be tight, the upper-level smoothing distribution prevents us from evaluating the base classifier on the entire matrix space. We leave the development of tight and efficient certificates for hierarchical randomized smoothing to future work.

**Gray-box certificates.** Scholten et al. (2022) derive gray-box certificates by exploiting the underlying message-passing principles of graph neural networks. In detail, they derive a tighter gray-box $\Delta$ that accounts for the fact that nodes may be disconnected from a target node when randomly deleting edges in a graph (and disconnected nodes do not affect the prediction for the target). Interestingly, we can directly plug their gray-box $\Delta$ into our framework to obtain gray-box robustness certificates for hierarchical randomized smoothing.

## H  Extended discussion

**Motivation for the indicator variable as additional column.** We certify robustness of classifiers operating on the higher-dimensional matrix space, which allows our certificates to be efficient and highly flexible. In the following we motivate this by considering the scenario of certifying robustness on the original data space $\mathcal{X}^{N \times D}$. The main problem is that it is technically challenging to obtain an efficient certificate that depends only on the radius (and not on fixed $X$ and $\tilde{X}$):

Consider a discrete lower-level smoothing distribution, fixed $X, \tilde{X}$, smoothed matrix $W$ and row $i \in \mathcal{C}$ such that $x_i \neq \tilde{x}_i$ but $x_i = w_i$. To certify robustness on the original space we have to consider the likelihood ratio for the marginal distribution $\Psi_X(W) \triangleq \sum_{\tau} \Psi_X(W, \tau)$:

$$\frac{\Psi_X(W)}{\Psi_{\tilde{X}}(W)} = \frac{\sum_{\tau} \Psi_X(W, \tau)}{\sum_{\tau} \Psi_{\tilde{X}}(W, \tau)}.$$

Let $\mathcal{C}$ describe all perturbed and $\tilde{\mathcal{C}}$ all clean rows. Similar due to independence we have:

$$\frac{\Psi_X(W)}{\Psi_{\tilde{X}}(W)} \overset{(1)}{=} \frac{\prod_{i=1}^n \Psi_{\mathbf{x}_i}(\mathbf{w}_i)}{\prod_{i=1}^n \Psi_{\tilde{\mathbf{x}}_i}(\mathbf{w}_i)} = \frac{\prod_{i\in\mathcal{C}} \Psi_{\mathbf{x}_i}(\mathbf{w}_i) \prod_{i\in\tilde{\mathcal{C}}} \Psi_{\mathbf{x}_i}(\mathbf{w}_i)}{\prod_{i\in\mathcal{C}} \Psi_{\tilde{\mathbf{x}}_i}(\mathbf{w}_i) \prod_{i\in\tilde{\mathcal{C}}} \Psi_{\tilde{\mathbf{x}}_i}(\mathbf{w}_i)} \overset{(2)}{=} \frac{\prod_{i\in\mathcal{C}} \Psi_{\mathbf{x}_i}(\mathbf{w}_i)}{\prod_{i\in\mathcal{C}} \Psi_{\tilde{\mathbf{x}}_i}(\mathbf{w}_i)}$$

where $(1)$ is due independence and $(2)$ due to $x_i = \tilde{x}_i$ for clean rows $i \in \tilde{\mathcal{C}}$. Now we can consider the perturbed row $i \in \mathcal{C}$ that we chose above and consider the likelihood ratio:

$$\frac{\Psi_{\mathbf{x}_i}(\mathbf{w}_i)}{\Psi_{\tilde{\mathbf{x}}_i}(\mathbf{w}_i)} = \frac{\sum_{\tau_i} \mu_{\mathbf{x}_i}(\mathbf{w}_i | \tau_i) \phi(\tau_i)}{\sum_{\tau_i} \mu_{\tilde{\mathbf{x}}_i}(\mathbf{w}_i | \tau_i) \phi(\tau_i)} \overset{(3)}{=} \frac{\mu_{\mathbf{x}_i}(\mathbf{w}_i) p + (1-p)}{p \mu_{\tilde{\mathbf{x}}_i}(\mathbf{w}_i)}$$

where $(3)$ is due to $w_i = x_i$ but $w_i \neq \tilde{x}_i$.

Due to the mixture, there are two ways sampling $w_i$ from $\Psi_{x_i}$: First, we may not "select" row $i$ (and then we do not smooth the row). Second, we may "select" row $i$ but not add any noise on the selected row. Due to the mixture, we cannot simply cancel out dimensions in which the rows $x_i$ and $\tilde{x}_i$ agree as in (Bojchevski et al., 2020), leading to the problem that the likelihood ratio and subsequently the point-wise certificate depends on $\tilde{X}$. Consequently, certifying robustness against an entire ball of perturbations is technically challenging since the likelihood ratio will change for every $\tilde{X}$.

We can resolve this problem by realizing that for reasonable smoothing parameters, the probability $\mu_{\tilde{x}_i}(w_i) = \mu_{\tilde{x}_i}(x_i)$ is close to zero (probability for flipping exactly *and only* the dimensions where $\tilde{x}_i$ disagrees with $x_i$ is close to zero for small radii and high dimensional data). Specifically, the likelihood ratio for such elements is approaching $\infty$ (e.g. for sparse smoothing with fixed radius $r_a, r_d$ we have $\mu_{\tilde{x}_i}(w_i) \to 0$ for $D \to \infty$). Note that if we choose a $W$ and row $i$ such that $x_i \neq \tilde{x}_i$ but $\tilde{x}_i = w_i$, we can derive the same but the other way around and find the likelihood ratio approaching $0$. In other words, we cannot avoid regions of $0$, and $\infty$ likelihood ratio for such smoothing distributions where we can flip "entire rows" from $x_i$ to $\tilde{x}_i$. Certifying robustness on the higher-dimensional matrix space instead is efficient, highly flexible and tight for higher dimensional rows as for example in graphs where rows represent node features.

**Efficiency of hierarchical randomized smoothing.** We discuss (1) the efficiency of the smoothing process, and (2) the efficiency of the certification. First, regarding the efficiency of the smoothing process, note that hierarchical smoothing only adds noise on a subset of all entities. Technically this can be more efficient, although drawing from a bernoulli before adding noise may require more time in practice than directly drawing (e.g. gaussian) noise for all matrix elements simultaneously. Second, regarding certification efficiency, our method only requires the additional computation of the term $\Delta$ which takes constant time. Yet, closed-form solutions for the largest certifiable radius may not be available, depending on the lower-level smoothing distribution. For example we may not be able to derive a closed-form radius for the multi-class certificate under hierarchical Gaussian smoothing (see Appendix E). Still, the certification process can be implemented efficiently even without closed-form solutions, e.g. via bisection or just by incrementally increasing $\epsilon$ to find the largest certifiable radius.

**Non-uniform entity-selection probability $p$.** In general we can also use a non-uniform $p$: Let $p_i$ denote a node-specific (non-uniform) selection probability for node $i$ that the adversary does not control. To get robustness certificates we have to make the worst-case assumption, i.e. nodes with lowest selection probability will be attacked first by the adversary. For the certificate this means we have to find the largest possible $\Delta$ (see also Theorem 1). Specifically, we have to choose $\Delta \triangleq 1 - \prod_{i \in \mathcal{I}} p_i$ where $\mathcal{I}$ contains the indices of the $r$ smallest probabilities $p_i$. Then we can proceed as in our paper to compute certificates for the smoothing distribution with non-uniform $p$. If we additionally know that certain nodes will be attacked before other nodes, we do not have to consider the first nodes with smallest $p_i$ but rather the first nodes with smallest $p_i$ under an attack order. This can be useful for example if one cluster of nodes is attacked before another, then we can increase the selection probability for nodes in the first cluster and obtain stronger certificates.

**Node-specific smoothing distributions.** If some nodes are exposed to different levels of attacks than others, we can obtain stronger certificates by making the lower-level smoothing distribution node-specific. This would allow us to add less noise for nodes that are less exposed to attacks, potentially leading to even better robustness-accuracy trade-offs. Note, however, that this increases threat model complexity since we will have to treat all nodes separately.

**Input-dependent randomized smoothing.** We derive certificates for a smoothing distribution that adds random noise independently on the matrix elements. This is the most studied class of smoothing distributions and allows us to integrate a whole suite of existing distributions. Although there are first approaches adding non-independent noise, the corresponding certificates are still in their infancy (Súkeník et al., 2022). Data-dependent smoothing is a research direction orthogonal to ours, and we consider the integration of such smoothing distributions as future work.

**Finding optimal smoothing parameters.** The randomized smoothing framework introduces parameters of the smoothing distribution as hyperparameters (see e.g. (Cohen et al., 2019)). In general, increasing the probabilities to add random noise increases the robustness guarantees but also decreases the accuracy. We introduce an additional parameter allowing to better control the robustness-accuracy trade-off. Specifically, larger $p$ adds more noise and increases the robustness but also decreases accuracy. In our exhaustive experiments we randomly explore the entire space of possible parameter combinations. The reason for this is to demonstrate Pareto-optimality, i.e. to find all ranges for which we can offer both better robustness and accuracy. In practice one would have to conduct significantly less experiments to find suitable parameters. Note that the task of efficiently finding the optimal parameters is a research direction orthogonal to deriving robustness certificates.

**Limitation of fixed threat models.** Recent adversarial attacks (see e.g. (Kollovieh et al., 2023)) go beyond the notion of $\ell_p$-norm balls and assess adversarial robustness using generative methods. Our approach is limited as we can only certify robustness against bounded adversaries that perturb at most $r$ objects with a perturbation strength that is bounded by $\epsilon$ under some $\ell_p$-norm. Since current robustness certificates can only provide provable guarantees under fixed threat models, certifying robustness against adversaries that are not bounded by $\ell_p$-norms is out of the scope of this paper.

**Graph-structure perturbations.** While we implement certificates against node-attribute perturbations, our framework can in principle integrate certificates against graph-structure perturbations (for example by bounding the number of nodes whose outgoing edges can be perturbed). This can be implemented by first selecting rows of the adjacency matrix $\boldsymbol{A}$ and then applying sparse smoothing (Bojchevski et al., 2020) to the selected rows only. Note that the works of Gao et al. (2020) and Wang et al. (2021) also study randomized smoothing against perturbations of the graph structure, but they consider a special case of sparse smoothing Bojchevski et al. (2020) with equal flipping probabilities.

