# OpenReview forum: "Hierarchical Randomized Smoothing"
_NeurIPS.cc/2023/Conference — NeurIPS 2023 poster_

### Official Review · Reviewer_BgsW · 2023-06-16

**Soundness:** 2 fair
**Presentation:** 2 fair
**Contribution:** 2 fair
**Rating:** 6
**Confidence:** 4

**Summary:**

The paper proposes a new threat model for the adversarial robustness - intersection of $\ell_0$ and $\ell_2$ ones where the $\ell_0$ is measured as the number of modified rows of a matrix. To certify robustness to this threat models, the authors propose a variant of randomized smoothing - hierarchical smoothing scheme. First, some rows are selected at random and then they are smoothed with Gaussian smoothing. Robustness certificate is provided for this kind of smoothing distribution and the superiority over baselines is demonstrated on some graph-network tasks.

**Strengths:**

* The paper proposes a new threat model that makes sense for modeling adversaries on graphs.
* The proposed certification method on this threat model (actually intersection of threat models) outperforms the baselines (methods designed for the individual threat models).

In general, this would be enough for me to vote for acceptance, however there are some problems as I list below.

**Weaknesses:**

* The technical novelty of the paper is limited; the paper focuses on a new threat model - an intersection of two standard threat models $\ell_0$ and $\ell_2$ - and the provided certification method is randomized smoothing where the smoothing distribution is essentially a product of the distribution used for $\ell_0$ (random mask) and $\ell_2$ (Gaussian noise). The certification is then analogical to (Cohen et al.).
* Please check all the claims if they actually say what you want to say. There are two main problems here - you consider a fixed $\hat{X}$ and provide certificate with respect to this particular point (e.g., in Coro 1 and also in the text.) whereas you obviously want to certify robustness to all points within this distance. And second, you assume that the perturbed point is at $\ell_0$ distance $r$ while it should be at a distance at most $r$ (e.g., lines 142-144 and from this point on).  These things should be very easy to fix to make it correct, but now it is just lax and wrong at places and it is the reason for the rejection at this point. **If this is fixed, then the paper becomes borderline acceptable for me**.
* The exposition is maybe unnecessarily complicated. The actual mathematical machinery performed in the paper boils down to decomposing the smoothing distribution as a mixture of two distributions (over $R_1, R_2$ or $R_2,R_3$ respectively in Propo 2) and then the NP lemma to two Gaussians is used for the distributions over $R_2$ (This is the standard NP-certification of (Cohen et al.)). It takes quite some effort to even understand "what is done" in the paper before I can focus on "why does it work". I suggest the authors to provide a pseudocode (e.g., as in (Cohen et al.)).

### minor
* The method has some parameters, but there is no suggestion on how to select them. They are selected randomly here and one can see a huge variance in the performance.
* The datasets are not explained. E.g., their dimensionality and rande of values. The discrete dataset contain some new parameters here but they are unexplained ($r_d, r_a, p_d, p_a..$)

**Questions:**

-

**Limitations:**

-

---

> ### Author Rebuttal · Authors · 2023-08-09
>
> Thank you for your review!
>
> Please note that we cannot update the paper during the rebuttal period according to the rebuttal policy, and we therefore carefully describe all changes directly here in the rebuttal.
>
> ### Concerning the technical contribution (Comment 1)
> Please consider that our certificates hold under adversarial perturbations that are bounded by arbitrary $\ell_p$-norms, unlike the result of Cohen et al., 2019 [1] that is limited to the $\ell_2$-norm. So far there are no randomized smoothing certificates specifically designed against our threat model and we experimentally demonstrate that simply using previous works is not enough when considering the robustness-accuracy trade-off. Deriving new certificates against this threat model is technically challenging: One would have to propose a new smoothing distribution and prove robustness certificates for each possible $\ell_p$-norm.
>
> As the result of a rigorous theoretical proof, we can provide robustness certificates that are more modular and in fact orthogonal to all existing ones. With hierarchical smoothing we also obtain stronger guarantees compared to the baselines. We believe that this is a sufficient contribution to the scientific community since it not only impacts the community of machine learning on graphs but also the (certifiable) robustness community in general.
>
> ### Concerning theoretical claims (Comment 2)
> Thank you for pointing out potential for using a more precise formulation in Section 4. In response to your comment, we carefully went over all theoretical statements to ensure their quality and further improve their clarity.
>
> Specifically, we clarified the statement regarding fixed $\tilde{\mathbf{X}}$ and a fix distance $r$ and rewrote Section 4 as follows: We first derive the NP-lower bound for a fixed point in the entire ball, yielding a point-wise certificate. Then we introduce guarantees against the entire threat model by including an additional proposition stating that the minimum of Theorem 1 (line 174) is assumed when exactly $r$ rows are perturbed (and not less). To see this note that $\Delta=1-p^{|\mathcal{C}|}$ is strictly monotonically increasing in the number of perturbed rows $|\mathcal{C}|$ for $p\in(0,1)$. We agree that this argument was missing, and we therefore fixed the corresponding parts. Thank you for helping us to further improve the clarity of our theoretical claims.
>
> Please further note that Corollary 1 is already correct and assumes any perturbed $\tilde{X}$ in the ball (line 207).
>
>
> ### Concerning the exposition (Comment 3)
> Thank you for helping us to further improve the exposition of our paper. In response to your comment, we simplified the explanations of the regions in Section 4 and additionally provide a Figure for clarification (see Figure 1 in the rebuttal PDF and also our response to reviewer 7yDS). Following your suggestion, we will also include pseudocode in the camera-ready version to further improve the clarity of our methodology.
>
> ### Concerning parameters of the smoothing distribution (Comment 4)
> The randomized smoothing framework already introduces parameters of the smoothing distribution as hyperparameters (see e.g. [1]). In general, increasing the probabilities to add noise increases the robustness guarantees but also decreases the accuracy. We introduce an additional parameter allowing to better control the robustness-accuracy trade-off.
>
> As you correctly pointed out, in our exhaustive experiments we randomly explore the entire space of possible parameter combinations. The reason for this is to demonstrate Pareto-optimality, i.e. to find all ranges for which we can offer both better robustness and accuracy. In practice one would have to conduct significantly less experiments to find suitable parameters. Please also consider that the task of efficiently finding the optimal parameters is a research direction orthogonal to deriving robustness certificates.
>
> ### Concerning further explanations (Comment 5)
> Please note that we thoroughly describe the graph datasets including number of nodes, edges and the feature dimensionality in lines 256-261 (Section 6). For experiments on discrete data we carefully introduce $r_d, r_a, p_d, p_a$ in lines 275-284 (Section 6) and we also elaborate on this in Appendix B.
>
> We hope that we could address all your questions to your satisfaction. Please let us know if you have any additional comments or questions.
>
> ### References
> [1] Jeremy M. Cohen, Elan Rosenfeld, J. Zico Kolter. Certified Adversarial Robustness via Randomized Smoothing. ICML 2019.

---

> > ### Comment · Reviewer_BgsW · 2023-08-15
> >
> > Thanks for the response.
> >
> > Regarding the generality of p-norms - I have read the paper two months ago, so I might not recall stuff precisely, but from what I remember, the method basically masks out something and then applies standard $\ell_p$ certification to the rest. Thus, I can use a smoothing distribution of my choice (and the certificates are straightforward as everything is just NP-lemma in one way or the other) in order to provide guarantees I want. If you do something more general than this, I would stress it out in the text.
> >
> > I did not realize that there is no updated pdf, so these changes would be good. With coro 1 I still have the problem that it is stated as a certificate for one point with some failure probability so in order to have a certificate for all the points uniformly, using this corollary I would need to union bound over all points that I clearly don't want; but clearly we can lower bound the probability once and that would work for all the points uniformly (which is exactly what everyone in randomized smoothing does), so I would consider rewriting coro 1 in the way that you don't fix the perturbation, but you state that the result holds for every perturbation.
> >
> > The $r_a, r_b, p_a, p_b$ thing - I checked the pdf again and I could find $r_a, r_b$, but could not find $p_a, p_b$. I am not familiar with the datasets so it is pretty hard to interpret the figures; maybe if there would be link from the captions to description/notation, it would be clearer.
> >
> > Anyway, thanks again for the response, I update me score and the things I wrote are some suggestions that you need not to reflect.

---

### Official Review · Reviewer_u4ou · 2023-07-06

**Soundness:** 2 fair
**Presentation:** 2 fair
**Contribution:** 2 fair
**Rating:** 4
**Confidence:** 2

**Summary:**

A randomized smoothing based robust certification approach is presented for machine learning under test-time/inference attacks, when only a subset of data is under attack e.g. a subset of nodes in graphical data. Robustness certificates are derived for discrete and continuous domains, and empirically certified and clean accuracies are computed and compared with prior work for certified robustness for graph neural networks (GNNs).

**Strengths:**

- The studied GNN model where only a subset of nodes of the graph may be attacked by the adversary is interesting and of practical significance.
- The proposed robust certification approach could achieve better trade-off of clean and certified accuracy than existing approaches.

**Weaknesses:**

- The overall approach seems like a simple extension to (Cohen et al. 2019), with a suitable adjustment factor $1-\Delta$ in the analysis corresponding to selecting a subset of rows. The theoretical analysis also follows similar arguments and it is not clear if any novel insights are obtained.
- A key tuning parameter for the approach is $p$, it is not clear how to set $p$ in a principled way.
- Missing several closely related prior research works e.g. [1] and [2].
- Results seem specific to bounded $\ell_2$ norm attacks.


[1] Wang, Binghui, Jinyuan Jia, Xiaoyu Cao, and Neil Zhenqiang Gong. "Certified robustness of graph neural networks against adversarial structural perturbation." In Proceedings of the 27th ACM SIGKDD Conference on Knowledge Discovery & Data Mining, pp. 1645-1653. 2021.
[2] Gao, Zhidong, Rui Hu, and Yanmin Gong. "Certified robustness of graph classification against topology attack with randomized smoothing." In GLOBECOM 2020-2020 IEEE Global Communications Conference, pp. 1-6. IEEE, 2020.

**Questions:**

- What is $\alpha$ in line 82? How is its value related to the certificates?
- Why is the noise applied independently to all matrix entries (e.g. line 97)?
- How do the theoretical results compare to Scholten et al. 2022?
- If $r=N$, i.e. all rows are under attack, how do current certificates compare to previously known certificates?

**Limitations:**

The authors could investigate further the limitations of the proposed approach and add a discussion.

---

> ### Author Rebuttal · Authors · 2023-08-09
>
> Thank you for your review!
>
> Please note that we cannot update the paper during the rebuttal period according to the rebuttal policy, and we therefore carefully describe all changes directly here in the rebuttal.
> ### Concerning the overall approach (Comment 1)
> So far there are no randomized smoothing certificates designed against our threat model and we demonstrate that simply using previous works is not enough when considering the robustness-accuracy trade-off. Deriving new certificates is technically challenging: One would have to propose a new smoothing distribution and prove robustness certificates for each $\ell_p$-norm. The modularity of our approach and the adjustment by $1-\Delta$ is the result of our particular smoothing distribution followed by a rigorous proof and represents a novel contribution: Our framework is of great flexibility since we can integrate existing certificates into our framework. We believe that our derivations represent an important contribution to the scientific community since our certificates provide better ways of assessing the robustness of GNNs and machine learning models in general.
> ### Concerning hyperparameter p (Comment 2)
> As you correctly pointed out, $p$ is a key parameter of our method and represents the probability of adding noise to the rows of a matrix. In response to your comment, we now provide additional intuition about $p$ in the paper: Specifically, larger $p$ adds more noise and increases the robustness but also decreases accuracy. By introducing this additional parameter we can better control the robustness-accuracy trade-off. Please consider that using hyperparameters of the smoothing distribution to control this trade-off is standard in randomized smoothing, and finding the optimal parameters efficiently is a research direction orthogonal to deriving robustness certificates.
> ### Concerning additional related work (Comment 3)
> In response to your comment, we revised the related work section and included the work [1] and [2] that you mentioned: The works of [1] and [2] derive robustness certificates against structural perturbations of the graph. While their certificates can be technically integrated into our framework, we experiment with the sparsity-preserving certificates proposed in [3] since it represents the current state-of-the-art in GNN certification.
> ### Concerning the threat model (Comment 4)
> Please note that our certificates hold under arbitrary $\ell_p$-norms (we just instantiate our framework with three different norms later in the experiments). In response to your comment, we rephrased the corresponding parts in the theoretical analysis (Section 4) for clarification.
> ### Concerning the parameter $\alpha$ (Question 1)
> Randomized smoothing certificates are probabilistic and $1-\alpha$ represents the confidence level used when estimating the smoothed classifier with Monte-Carlo samples. Please note that this is an inherent property of probabilistic certification and not a limitation of our work specifically. In response to your comment, we added further clarifications in the background section when introducing the randomized smoothing framework.
> ### Concerning independent noise (Question 2)
> As you correctly pointed out, we derive certificates for a smoothing distribution that adds random noise independently on the matrix elements. This is the most studied class of smoothing distributions and allows us to integrate a whole suite of existing distributions. Although there are first approaches adding non-independent noise, the corresponding certificates are still in their infancy [4]. Please note that non-independent noise is a research direction orthogonal to ours. Integrating such smoothing distributions is an interesting idea for future work.
> ### Concerning the theoretical comparison (Question 3)
> The certificate proposed in [5] is based on ablation smoothing and can be considered as a special case of our framework. In contrast, our paper represents a novel contribution that goes beyond simple ablation: We first select rows but instead of ablating them we add random noise to them. As we show experimentally, our method yields stronger results under our threat model when compared to [5].
> ### Concerning the special case $r=N$ (Question 4)
> For $r=N$ (all rows are under attack) we recover the threat model already studied in the literature (see e.g. [3]). In this special case our certificates are exactly as strong as existing methods (see lines 195-201). Please consider that we are proposing novel certificates specifically designed against the threat model of $r \ll N$ (multiple but not all rows are under attack), which is a more realistic assumption for graphs and already actively exploited in several adversarial attacks against GNNs [6].
> ### Concerning the discussion (Comment 5)
> Please consider that we already discuss the limitations of our approach in Section 5. In response to your review, we further extended the discussion: Specifically, we included the hyperparameter discussion and the non-independent noise discussion (see our response to Comment 2 and Question 2). Thank you for helping us to further improve the paper.
>
> We hope that we could address all your questions to your satisfaction. Please let us know if you have any additional comments or questions.
> ### References
> [3] Aleksandar Bojchevski, Johannes Klicpera, Stephan Günnemann: Efficient Robustness Certificates for Discrete Data. Sparsity-Aware Randomized Smoothing for Graphs, Images and More. ICML 2020.
>
> [4] Peter Súkeník, Aleksei Kuvshinov, Stephan Günnemann. Intriguing Properties of Input-Dependent Randomized Smoothing. ICML 2022.
>
> [5] Yan Scholten, Jan Schuchardt, Simon Geisler, Aleksandar Bojchevski, Stephan Günnemann. Randomized Message-Interception Smoothing: Gray-box Certificates for Graph Neural Networks. NeurIPS 2022.
>
> [6] Jiaqi Ma, Shuangrui Ding, Qiaozhu Mei. Towards More Practical Adversarial Attacks on Graph Neural Networks. NeurIPS 2020.

---

> > ### Comment · Reviewer_u4ou · 2023-08-14
> > **Reply to rebuttal**
> >
> > Thank you for the detailed response. In particular, thanks for providing the clarifications for my questions.
> >
> > Overall, there are a lot of edits needed to improve the readability of the theoretical results and make the paper ready for publication in my opinion. Also further related work needs to be discussed (e.g. [4] and [6] in the rebuttal) and compared with. This is particularly important since the work proposes a new attack model for GNNs.

---

> > > ### Author Response · Authors · 2023-08-16
> > > **Response to Reviewer u4ou**
> > >
> > > Thank you for your response!
> > >
> > > For the revised version of our manuscript, we would be grateful if you could clarify which further edits are needed based on your original review.
> > >
> > > In response to your questions, we suggested in our rebuttal:
> > > - Adding a few clarifications regarding the background of randomized smoothing, and
> > > - Including the related work from our discussion.
> > >
> > > Regarding the works [4] and [6], we are happy to include them in the related work section. However, it is important to point out that these are orthogonal to our core research question. Specifically, [4] represents a negative result concerning input-dependent randomized smoothing (which we do not work on), and [6] proposes an adversarial attack. In contrast, we develop novel robustness certificates.

---

### Official Review · Reviewer_7yDS · 2023-07-06

**Soundness:** 4 excellent
**Presentation:** 3 good
**Contribution:** 3 good
**Rating:** 7
**Confidence:** 4

**Summary:**

The paper introduces a new variant of the randomized smoothing algorithm for certified robustness. The paper considers the setting where the input data is split into multiple parts or sites (e.g., a graph) and an adversary can perturb at most $r$ sites at the same time.
To obtain robustness certificates for this setting, the authors introduce hierarchical randomized smoothing.

In standard randomized smoothing (Gaussian) noise is added to the input, a classification model is evoked on the noisy input, and the lost likely output class is taken as the final prediction.
In practice this is done via Monte Carlo estimate: Noise is sampled multiple times and the majority vote is returned.

Hierarchical randomized smoothing mimics this process but splits the sampling of the noise into two steps: i) first a random indicator variable determining the affected sites is sampled, ii) noise is added to those sites as before.

To obtain a guarantee in this setting, the theoretical derivation combines the standard form for randomized smoothing for continuous (Gaussian) with a variant for discrete noise. Ultimately, this approach recovers the same certificate as the standard randomized smoothing algorithm, up to a scaling factor depending on the number of sites that can be perturbed at the same time, thus allowing for improvements.


**Strengths:**

- Strong and original contribution
- Well written
- Rigorous mathematical derivation
- Good empirical results


**Weaknesses:**

I have no major concerns, other than maybe the applicability of the threat model considered in the paper (see questions).
However, I have some minor comments mostly on the presentation of the paper:

- In the example around line 153, I found the set $\mathcal{R}_0$ confusing as the explanation only comes on the next page.
- Similarly the construction of Propositions 1 and 2 might be easier to follow if it was visualized.

**Questions:**

- Does row selection $p$ need to be uniform? What is the impact on the certificate if it is used to encode a prior on the likelihood of change/attack for different part.s
- Can you further comment on the threat model? Not mathematically, but rather on the settings in which the proposed combination of LP-constraints over $r$ site changes is applicable.

**Limitations:**

Limitations and broader impact are adequately discussed in the paper.

---

> ### Author Rebuttal · Authors · 2023-08-09
>
> Thank you for your review!
>
> Please note that we cannot update the paper during the rebuttal period according to the rebuttal policy, and we therefore carefully describe all changes directly here in the rebuttal.
>
> ### Concerning a visualization of Propositions 1 and 2 (Comment 1 and 2)
> Thank you for your suggestion to further improve the clarity of our claims. In response to your comment, we rewrote Section 4 to make our derivations more accessible. In this context we also included a new Figure to visualize the regions and the propositions (see rebuttal PDF in the global response). We also followed your suggestion and explain the regions and $\mathcal{R}_0$ earlier.
>
> ### Concerning row-selection probability $p$ (Question 1)
> In our paper we currently select rows independently from each other with a row-selection probability $p$. Although there are first non-hierarchical smoothing distributions adding data-dependent noise, the corresponding certificates are still in their infancy (see e.g. [1]). The challenge of making our hierarchical smoothing data-dependent is that using higher $p$ for some parts of the input means adversaries could simply attack the other parts. Please consider that deriving robustness certificates for data-dependent smoothing distributions represents a research direction orthogonal to ours. Thank you for pointing us to this interesting idea for future work.
>
> ### Concerning further applications for our threat model (Question 2)
>
> As you correctly pointed out, we consider the threat model where adversaries can perturb at most $r$ entities of an object. This is a realistic assumption for graphs such as social networks and already actively exploited in several recent adversarial attacks against graph neural networks (see e.g. [2]). For example, consider you want to attack advertisements in Facebook: Probably you will not be able to control the entire Facebook graph, but you may be able to buy a few hundred accounts.
>
> Since we propose certificates for data where objects can be decomposed into multiple entities there are numerous applications beyond graphs: For example in Natural Language Processing, adversaries may be restricted to perturb at most $r$ documents in a collection, or at most $r$ paragraphs in a document. Further real-world examples include applications where reliability and security are crucial, this includes especially the medical and financial domains. Since graphs are ubiquitous datastructures the applicability of our certificates is versatile with a far-reaching impact.
>
> Beyond their real-life applicability, robustness certificates are also generally useful tools that allow us to assess the robustness of models beyond adversarial perturbations (for example against noisy signals or incomplete data). Robustness certificates can also provide insights to design more robust models in the future.
>
> We hope that we could address all your questions to your satisfaction. Please let us know if you have any additional comments or questions.
>
> ### References
> [1] Peter Súkeník, Aleksei Kuvshinov, Stephan Günnemann. Intriguing Properties of Input-Dependent Randomized Smoothing. ICML 2022.
>
> [2] Jiaqi Ma, Shuangrui Ding, Qiaozhu Mei. Towards More Practical Adversarial Attacks on Graph Neural Networks. NeurIPS 2020.

---

> > ### Comment · Reviewer_7yDS · 2023-08-11
> > **Reply**
> >
> > I thank the authors for their rebuttal. I greatly appreciate the new Figure.
> >
> > With regards to Question 1: My suggestion was not (necessarily) to make the distribution data-dependent, which I few as quite problematic, but rather have a non-uniform a-priory p for all notes depending on some key property of the graph/nodes that the adversary does not control. Especially in a distributed setting (as might be realistic for many of the discussed scenarios), certain nodes may depend more or less on external input and therefore might be exposed to differing levels of attacks.

---

> > > ### Author Response · Authors · 2023-08-14
> > > **Response to Reviewer 7yDS**
> > >
> > > Thank you for your response!
> > >
> > > Regarding question 1 $-$ In general we can use a non-uniform $p$:
> > >
> > > Let $p_i$ denote a node-specific (non-uniform) selection probability for node $i$ that the adversary does not control. To get robustness certificates we have to make the worst-case assumption, i.e. the nodes with lowest selection probability will be attacked first by the adversary. For the certificate this means we have to find the largest possible $\Delta$ (compare Theorem 1). Specifically, we have to choose $\Delta = 1-\prod_{i\in\mathcal{I}} p_i$ where $\mathcal{I}$ contains the indices $i$ of the $r$ smallest probabilities $p_i$. Then we can proceed as in our paper to compute certificates for the smoothing distribution with non-uniform $p$.
> > >
> > > If we additionally know that certain nodes will be attacked before other nodes, we do not have to consider the first $r$ nodes with smallest $p_i$ but rather the first $r$ nodes with smallest $p_i$ under an attack order. This can be useful for example if one cluster of nodes is attacked before another, then we can increase the selection probability for nodes in the first cluster and obtain stronger certificates.
> > >
> > > As you suggested in your response, some nodes might be exposed to different levels of attacks than others. Here, making the lower-level smoothing distribution node-specific might also be beneficial: This would allow us to add less noise for nodes that are less exposed to attacks, potentially leading to even better robustness-accuracy trade-offs.
> > >
> > > We will add this discussion to the paper. Please let us know if you have any additional comments or questions.

---

### Official Review · Reviewer_FfTs · 2023-07-07

**Soundness:** 3 good
**Presentation:** 3 good
**Contribution:** 3 good
**Rating:** 6
**Confidence:** 4

**Summary:**

This paper proposes hierarchical randomized smoothing, a variant of randomized smoothing that not only randomly perturbs input data at test time but also randomly selects which rows of the input to perturb. This is motivated by a threat model in which an adversary only selects a subset of rows of matrix-valued data to attack. The authors show that their hierarchical smoothing scheme inherits certified robustness by the lower-level smoothing scheme used (i.e., the randomization method on the rows to be perturbed), and also show that the hierarchical scheme generalizes randomized ablation (random masking of elements rather than random additive noise). Experiments are conducted on benchmark node classification datasets that show the increased certification power over prior smoothing methods that do not account for the row-selection limitations of the adversary.

**Strengths:**

1. The paper is easy to follow, well-written, and appears to have some nice novel ideas for exploiting the structure of adversarial attacks on graph neural networks and related "hierarchical" models in order to get tighter robustness certificates than structure-blind methods.

2. It is interesting that the proposed approach generalizes both conventional additive noise certificates as well as ablation (masking) certificates.

3. The proposed hierarchical approach seems to be quite modular, allowing for new lower-level certification methods to be incorporated as they are developed in the future.

**Weaknesses:**

See below "Questions."

**Questions:**

1. In my opinion, the overarching structure of the paper could be improved. I'd recommend moving the Discussion section to the end of the paper with the Conclusion, and also moving at least the more "general" parts of the Related Work section to the end of the Introduction.

2. Line 276: "discrete, binary features: $\mathcal{X} = [0,1]$." Do you mean $\mathcal{X} = \\{0,1\\}$?

3. Line 302: Should "(2) evaluating certificates" be "(3)"?

4. The experimental comparisons seem a bit misleading. In particular, you are not making an entirely fair comparison to conventional RS, since those certificates are tailored to threat models where the adversary has access to perturb every entry in your input data. It is therefore no surprise that your certified radii are larger, since you are considering a weaker adversary altogether. How do your certificates perform in the extreme case where the adversary does have access to all entries in the matrix? Do your certificates still outperform conventional RS?

**Limitations:**

N/A.

---

> ### Author Rebuttal · Authors · 2023-08-09
>
> Thank you for your review!
>
> Please note that we cannot update the paper during the rebuttal period according to the rebuttal policy, and we therefore carefully describe all changes directly here in the rebuttal.
>
> ### Concerning the paper structure (Question 1)
> We followed your suggestion and moved the (improved) related work after the introduction, and the discussion to the end. We found that this way we also better highlight the novelty of our contributions and the contrast to related work. Thank you for helping us to further improve the paper.
>
> ### Concerning the typos in line 276 and 302 (Question 2 and 3)
> Yes you are right, we fixed the corresponding lines.
>
> ### Concerning the experimental comparison (Question 4)
> Since our certificates generalize ablation and additive noise certificates into a new common framework, our method performs either better or on-par with the existing methods. To see this, for $r=N$ (all rows are under attack) we recover the threat model already studied in the literature (see e.g. [1,2]). In the special case of choosing $p=1$, our certificate boils down to the standard certification methods and therefore performs on-par with the baselines (see discussion in lines 195-201).
>
> Please consider that the threat model where adversaries can only attack a subset of all rows ($r \ll N$) is a more realistic assumption for graphs such as social networks and already actively exploited in several recent adversarial attacks against graph neural networks (see e.g. [3]). Since we propose the first smoothing distribution specifically tailored against the described threat model, we believe our experimental comparison to existing certificates is fair and scientifically sound.
>
> We hope that we could address all your questions to your satisfaction. Please let us know if you have any additional comments or questions.
>
> ### References
> [1] Jeremy M. Cohen, Elan Rosenfeld, J. Zico Kolter. Certified Adversarial Robustness via Randomized Smoothing. ICML 2019.
>
> [2] Aleksandar Bojchevski, Johannes Klicpera, Stephan Günnemann: Efficient Robustness Certificates for Discrete Data. Sparsity-Aware Randomized Smoothing for Graphs, Images and More. ICML 2020.
>
> [3] Jiaqi Ma, Shuangrui Ding, Qiaozhu Mei. Towards More Practical Adversarial Attacks on Graph Neural Networks. NeurIPS 2020.

---

> > ### Comment · Reviewer_FfTs · 2023-08-12
> >
> > I thank the authors for their responses and updates to the paper. To better motivate your threat model, you should consider citing [3] in the paper, which I did not see in the original submission.
> >
> > Given the unsurprising comparison to conventional RS (which is designed for a more general threat model), I maintain my original score. Are there no other certification techniques (perhaps even non-smoothing based ones) that directly consider your specific threat model?

---

> > > ### Author Response · Authors · 2023-08-14
> > > **Response to Reviewer FfTs**
> > >
> > > Thank you for your response, we will add the citation to the revised paper.
> > >
> > > So far there are no robustness certificates specifically designed against our threat model, which again motivates the need for our certificates and the novelty of our method. Please consider that there are also no non-smoothing certificates against our threat model. We therefore demonstrate Pareto-optimality of our method when compared to the three existing state-of-the-art baselines for robustness certification of graph neural networks.

---

### Author Rebuttal · Authors · 2023-08-09

We would like to thank all reviewers for their valuable feedback.

Please note that we cannot update the paper during the rebuttal period according to the rebuttal policy, and we therefore carefully describe all revisions directly in the rebuttals. In the rebuttal PDF we further include a Figure as clarification of the regions and propositions as suggested by the reviewers (see PDF attached).

---

### Decision · Program_Chairs · 2023-09-21

**Decision:**

Accept (poster)

**Comment:**

I have thoroughly read the reviews, rebuttals, and subsequent discussions. The paper has received generally positive initial scores, and it's encouraging to note that several reviewers have further improved their evaluations after considering the rebuttal.

The paper introduces a novel approach that addresses a new type of attacks on structured data, such as graphs. The theoretical results, while potentially having limitations in their applicability, generalize existing randomized smoothing (RS) certificates for when a subset of size "r" of the nodes is perturbed (each node with a probability of perturbation = p). The certificates are now more general and function of both p and r. It would be intriguing to explore an extension of this work for scenarios where the adversary can perturb at most "r" nodes, as opposed to exactly "r" nodes. This aligns more closely with realistic attack scenarios, where one knows a maximum adversarial budget rather than the exact adversarial budget.

Nonetheless, the contributions and experiments presented in the paper are robust and well-founded. Based on my evaluation, I recommend accepting the paper, in alignment with the sentiments of all reviewers.